# Bridging Imitation and Online Reinforcement Learning: An Optimistic Tale

**Botao Hao**                                                     *bhao@google.com*
*Google DeepMind*

**Rahul Jain**                                                 *rahul.jain@usc.edu*
*University of Southern California*
*Google DeepMind**

**Dengwang Tang**                                              *dengwang@usc.edu*
*University of Southern California*

**Zheng Wen**                                                 *zhengwen@google.com*
*Google DeepMind*

**Reviewed on OpenReview:** *https://openreview.net/forum?id=lanGfXOM6C*

## Abstract

In this paper, we address the following problem: Given an offline demonstration dataset from an imperfect expert, what is the best way to leverage it to bootstrap online learning performance in MDPs. We first propose an Informed Posterior Sampling-based RL (iPSRL) algorithm that uses the offline dataset, and information about the expert's behavioral policy used to generate the offline dataset. Its cumulative Bayesian regret goes down to zero exponentially fast in $N$, the offline dataset size if the expert is competent enough. Since this algorithm is computationally impractical, we then propose the iRLSVI algorithm that can be seen as a combination of the RLSVI algorithm for online RL, and imitation learning. Our empirical results show that the proposed iRLSVI algorithm is able to achieve significant reduction in regret as compared to two baselines: no offline data, and offline dataset but used without suitably modeling the generative policy. Our algorithm can be seen as bridging online RL and imitation learning.

## 1 Introduction

An early vision of the Reinforcement Learning (RL) field is to design a learning agent that when let loose in an unknown environment, learns by interacting with it. Such an agent starts with a blank slate (with possibly, arbitrary initialization), takes actions, receives state and reward observations, and thus learns by "reinforcement". This remains a goal but at the same time, it is recognized that in this paradigm learning is too slow, inefficient and often impractical. Such a learning agent takes too long to learn near-optimal policies way beyond practical time horizons of interest. Furthermore, deploying an agent that learns by exploration over long time periods may simply be impractical.

In fact, reinforcement learning is often deployed to solve complicated engineering problems by first collecting offline data using a behavioral policy, and then using off-policy reinforcement learning, or imitation learning methods (if the goal is to imitate the policy that generated the offline dataset) on such datasets to learn a policy. This often suffers from the *distribution-shift* problem, i.e., the learnt policy upon deployment often performs poorly on out-of-distribution state-action space. Thus, there is a need for adaptation and fine-tuning upon deployment.

---

*Work done while at Google DeepMind.

In this paper, we propose a systematic way to use offline datasets to bootstrap online RL algorithms. Performance of online learning agents is often measured in terms of cumulative (expected) regret. We show that, as expected, there is a gain in performance (reflected in reduction in cumulative regret) of the learning agent as compared to when it did not use such an offline dataset. We call such an online learning agent as being *partially informed*. However, somewhat surprisingly, if the agent is further informed about the behavioral policy that generated the offline dataset, such an *informed (online learning) agent* can do substantially better, reducing cumulative regret significantly. In fact, we also show that if the behavioral policy is suitably parameterized by a *competence parameter*, wherein the behavioral policy is asymptotically the optimal policy, then the higher the "competence" level, the better the performance in terms of regret reduction over the baseline case of no offline dataset.

We first propose an ideal (informed) `iPSRL` (posterior sampling-based RL) algorithm and show via theoretical analysis that under some mild assumptions, its expected cumulative regret is bounded as $\tilde{O}(\sqrt{T})$ where $T$ is the number of episodes. In fact, we show that if the competence of the expert is high enough (quantified in terms of a parameter we introduce), the regret goes to zero exponentially fast as $N$, the offline dataset size grows. This is accomplished through a novel prior-dependent regret analysis of the PSRL algorithm, the first such result to the best of our knowledge. Unfortunately, posterior updates in this algorithm can be computationally impractical. Thus, we introduce a Bayesian-bootstrapped algorithm for approximate posterior sampling, called the (informed) `iRLSVI` algorithm (due to its commonality with the RLSVI algorithm introduced in Osband et al. (2019)). The `iRLSVI` algorithm involves optimizing a loss function that is an *optimistic* upper bound on the loss function for MAP estimates for the unknown parameters. Thus, while inspired by the posterior sampling principle, it also has an optimism flavor to it. Through, numerical experiments, we show that the `iRLSVI` algorithm performs substantially better than both the partially informed-RLSVI (which uses the offline dataset naively) as well as the uninformed-RLSVI algorithm (which doesn't use it at all).

We also show that the `iRLSVI` algorithm can be seen as bridging online reinforcement learning with imitation learning since its loss function can be seen as a combination of an online learning term as well as an imitation learning term. And if there is no offline dataset, it essentially behaves like an online RL algorithm. Of course, in various regimes in the middle it is able to interpolate seamlessly.

**Related Work.** Because of the surging use of offline datasets for pre-training (e.g., in Large Language models (LLMs), e.g., see Brown et al. (2020); Thoppilan et al. (2022); Hoffmann et al. (2022)), there has been a lot of interest in Offline RL, i.e., RL using offline datasets (Levine et al., 2020). A fundamental issue this literature addresses is RL algorithm design (Nair et al., 2020; Kostrikov et al., 2021; Kumar et al., 2020; Nguyen-Tang & Arora, 2023; Fujimoto et al., 2019; Fujimoto & Gu, 2021; Ghosh et al., 2022) and analysis to best address the "out-of-distribution" (OOD) problem, i.e., policies learnt from offline datasets may not perform so well upon deployment. The dominant design approach is based on 'pessimism' (Jin et al., 2021; Xie et al., 2021a; Rashidinejad et al., 2021) which often results in conservative performance in practice. Some of the theoretical literature (Xie et al., 2021a; Rashidinejad et al., 2021; Uehara & Sun, 2021; Agarwal & Zhang, 2022) has focused on investigation of sufficient conditions such as "concentrability measures" under which such offline RL algorithms can have guaranteed performance. Unfortunately, such measures of offline dataset quality are hard to compute, and of limited practical relevance (Argenson & Dulac-Arnold, 2020; Nair et al., 2020; Kumar et al., 2020; Levine et al., 2020; Kostrikov et al., 2021; Wagenmaker & Pacchiano, 2022).

There is of course, a large body of literature on online RL (Dann et al., 2021; Tiapkin et al., 2022; Ecoffet et al., 2021; Guo et al., 2022; Ecoffet et al., 2019; Osband et al., 2019) with two dominant design philosophies: Optimism-based algorithms such as UCRL2 in Auer et al. (2008), and Posterior Sampling (PS)-type algorithms such as PSRL (Osband et al., 2013; Ouyang et al., 2017), etc. (Osband et al., 2016a;b; 2019; Russo & Van Roy, 2018; Zanette & Sarkar, 2017; Wen et al., 2020; Hao & Lattimore, 2022). However, none of these algorithms consider starting the learning agent with an offline dataset. Of course, imitation learning (Hester et al., 2018; Beliaev et al., 2022; Schaal, 1996) is exactly concerned with learning the expert's behavioral policy (which may not be optimal) from the offline datasets but with no online finetuning of the policy learnt. Several papers have actually studied bridging offline RL and imitation learning (Ernst et al., 2005; Kumar et al., 2022; Rashidinejad et al., 2021; Hansen et al., 2022; Vecerik et al., 2017; Lee et al., 2022). Some have

also studied offline RL followed by a small amount of policy fine-tuning (Song et al., 2022; Fang et al., 2022; Xie et al., 2021b; Wan et al., 2022; Schrittwieser et al., 2021; Ball et al., 2023; Uehara & Sun, 2021; Xie et al., 2021b; Agarwal & Zhang, 2022) with the goal of finding policies that optimize simple regret.

The problem we study in this paper is motivated by a similar question: Namely, given an offline demonstration dataset from an imperfect expert, what is the best way to leverage it to bootstrap online learning performance in MDPs? However, our work is different in that it focuses on cumulative regret as a measure of online learning performance. This requires smart exploration strategies while making maximal use of the offline dataset to achieve the best regret reduction possible over the case when an offline dataset is not available. Of course, this will depend on the quality and quantity of the demonstrations. And the question is what kind of algorithms can one devise to achieve this objective, and what information about the offline dataset-generation process is helpful? What is the best regret reduction that is achievable by use of offline datasets? How it depends on the quality and quantity of demonstrations, and what algorithms can one devise to achieve them? And does any information about the offline-dataset generation process help in regret reduction? We answer some of these questions in this paper.

## 2 Preliminaries

**Episodic Reinforcement Learning.** Consider a scenario where an agent repeatedly interacts with an environment modelled as a finite-horizon MDP, and refer to each interaction as an episode. The finite-horizon MDP is represented by a tuple $\mathcal{M} = (\mathcal{S}, \mathcal{A}, P, r, H, \nu)$, where $\mathcal{S}$ is a finite state space (of size $S$), $\mathcal{A}$ is a finite action space (of size $A$), $P$ encodes the transition probabilities, $r$ is the reward function, $H$ is the time horizon length, and $\nu$ is the initial state distribution. The interaction protocol is as follows: at the beginning of each episode $t$, the initial state $s_0^t$ is independently drawn from $\nu$. Then, at each period $h = 0, 1, \ldots, H-1$ in episode $t$, if the agent takes action $a_h^t \in \mathcal{A}$ at the current state $s_h^t \in \mathcal{S}$, then it will receive a reward $r_h(s_h^t, a_h^t)$ and transit to the next state $s_{h+1}^t \in P_h(\cdot|s_h^t, a_h^t)$. An episode terminates once the agent arrives at state $s_H^t$ in period $H$ and receives a reward $r_H(s_H^t)$. We abuse notation for the sake of simplicity, and just use $r_H(s_H^t, a_H^t)$ instead of $r_H(s_H^t)$, though no action is taken at period $H$. The objective is to maximize its expected total reward over $T$ episodes.

Let $Q_h^*$ and $V_h^*$ respectively denote the optimal state-action value and state value functions at period $h$. Then, the Bellman equation for MDP $\mathcal{M}$ is

$$Q_h^*(s, a) = r_h(s, a) + \sum_{s'} P_h(s'|s, a) V_{h+1}^*(s'), \tag{1}$$

where $V_{h+1}^*(s') := \max_b Q_{h+1}^*(s', b)$, if $h < H-1$ and $V_{h+1}^*(s') = 0$, if $h = H-1$. We define a policy $\pi$ as a mapping from a state-period pair to a probability distribution over the action space $A$. A policy $\pi^*$ is optimal if $\pi_h^*(\cdot|s) \in \arg\max_{\pi_h} \sum_a Q_h^*(s, a) \pi_h(a|s)$ for all $s \in \mathcal{S}$ and all $h$.

**Agent's Prior Knowledge about $\mathcal{M}$.** We assume that the agent does not fully know the environment $\mathcal{M}$; otherwise, there is no need for learning and this problem reduces to an optimization problem. However, the agent usually has some prior knowledge about the unknown part of $\mathcal{M}$. For instance, the agent might know that $\mathcal{M}$ lies in a low-dimensional subspace, or may have a prior distribution over $\mathcal{M}$. We use the notation $\mathcal{M}(\theta)$ where $\theta$ parameterizes the unknown part of the MDP. When we want to emphasize it as a random quantity, we will denote it by $\theta^*$ with a prior distribution $\mu_0$. Of course, different assumptions about the agent's prior knowledge lead to different problem formulations and algorithm designs. As a first step, we consider two canonical settings:

- **Tabular RL:** The agent knows $\mathcal{S}, \mathcal{A}, r, H$ and $\nu$, but does not know $P$. That is, $\theta^* = P$ in this setting. We also assume that the agent has a prior over $P$, and this prior is independent across state-period-action triples.

- **Linear value function generalization:** The agent knows $\mathcal{S}, \mathcal{A}, H$ and $\nu$, but does not know $P$ and $r$. Moreover, the agent knows that for all $h$, $Q_h^*$ lies in a low-dimensional subspace $\text{span}(\Phi_h)$,

where $\Phi_h \in \Re^{|S||A| \times d}$ is a known matrix. In other words, $Q_h^* = \Phi_h \theta_h^*$ for some $\theta_h^* \in \Re^d$. Thus, in this setting $\theta^* = \left[ \theta_0^{*\top}, \ldots, \theta_{H-1}^{*\top} \right]^{\top}$. We also assume that the agent has a Gaussian prior over $\theta^*$.

As we will discuss later, the insights developed in this paper could potentially be extended to more general cases.

**Offline Datasets.** We denote an *offline dataset* with $L$ episodes as $\mathcal{D}_0 = \{(\bar{s}_0^l, \bar{a}_0^l, \cdots, \bar{s}_H^l)_{l=1}^L\}$, where $N = HL$ denotes the dataset size in terms of number of observed transitions. For the sake of simplicity, we assume we have complete trajectories in the dataset but it can easily be generalized if not. We denote an *online dataset* with $t$ episodes as $\mathcal{H}_t = \{(s_0^l, a_0^l, \cdots, s_H^l)_{l=1}^t\}$ and $\mathcal{D}_t = \mathcal{D}_0 \oplus \mathcal{H}_t$.

**The Notion of Regret.** A online learning algorithm $\phi$ is a map for each episode $t$, and time $h$, $\phi_{t,h} : \mathcal{D}_t \to \Delta_A$, the probability simplex over actions. We define the Bayesian regret of an online learning algorithm $\phi$ over $T$ episodes as

$$\mathfrak{BR}_T(\phi) := \mathbb{E}\left[\sum_{t=1}^T \left( V_0^*(s_0^t; \theta^*) - \sum_{h=0}^H r_h(s_h^t, a_h^t) \right)\right],$$

where the $(s_h^t, a_h^t)$'s are the state-action tuples from using the learning algorithm $\phi$, and the expectation is over the sequence induced by the interaction of the learning algorithm and the environment, the prior distributions over the unknown parameters $\theta^*$ and the offline dataset $\mathcal{D}_0$.

**Expert's behavioral policy and competence.** We assume that the expert that generated the offline demonstrations may not be perfect, i.e., the actions it takes are only approximately optimal with respect to the optimal $Q$-value function. To that end, we model the expert's policy by use of the following generative model,

$$\pi_h^\beta(a|s) = \frac{\exp(\beta(s) Q_h^*(s,a))}{\sum_a \exp(\beta(s) Q_h^*(s,a))}, \tag{2}$$

where $\beta(s) \geq 0$ is called the *state-dependent deliberateness* parameter, e.g., when $\beta(s) = 0$, the expert behaves naively in state $s$, and takes actions uniformly randomly. When $\beta(s) \to \infty$, the expert uses the optimal policy when in state $s$. When $\beta(\cdot)$ is unknown, we will assume an independent exponential prior for the sake of analytical simplicity, $f_2(\beta(s)) = \lambda_2 \exp(-\lambda_2 \beta(s))$ over $\beta(s)$ where $\lambda_2 > 0$ is the same for all $s$. In our experiments, we will regard $\beta(s)$ as being the same for all states, and hence a single parameter.

The above assumes the expert is knowledgeable about $Q^*$. However, it may know it only approximately. To model that, we introduce a *knowledgeability* parameter $\lambda \geq 0$. The expert then knows $\tilde{Q}$ which is distributed as $\mathcal{N}(Q^*, \mathbb{I}/\lambda^2)$ conditioned on $\theta$, and selects actions according to the softmax policy Eq. (2), with the $Q^*$ replaced by $\tilde{Q}$. The two parameters $(\beta, \lambda)$ together will be referred to as the *competence* of the expert. In this case, we denote the expert's policy as $\pi_h^{\beta,\lambda}$.

*Remark* 2.1. While the form of the generative policy in Eq. (2) seems specific, $\pi_h^\beta(\cdot|s)$ is a random vector with support over the entire probability simplex. In particular, if one regards $\beta(s)$ and $\tilde{Q}_h(s,\cdot)$ as parameters that parameterize the policy, the softmax policy structure as in Eq. (2) is enough to realize any stationary policy.

Furthermore, we note that our main objective here is to yield clear and useful insights when information is available to be able to model the expert's behavioral policy with varying competence levels. Other forms of generative policies can also be used including $\epsilon$-optimal policies introduced in (Beliaev et al., 2022), and the framework extended.

## 3  The Informed PSRL Algorithm

We now introduce a simple *Informed Posterior Sampling-based Reinforcement Learning* (`iPSRL`) algorithm that naturally uses the offline dataset $\mathcal{D}_0$ and action generation information to construct an informed prior distribution over $\theta^*$. The realization of $\theta^*$ is assumed known to the expert (but not the learning agent) with

---

**Algorithm 1** iPSRL

---

**Input:** Prior $\mu_0$, Initial state distribution $\nu$
**for** $t = 1, \cdots, T$ **do**

   (A1) Update posterior, $\mu_t(\theta|\mathcal{H}_{t-1}, \mathcal{D}_0)$ using Bayes' rule

   (A2) Sample $\tilde{\theta}_t \sim \mu_t$

   Compute optimal policy $\tilde{\pi}^t(\cdot|s) := \pi_h^*(\cdot|s; \tilde{\theta}_t)$ (for all $h$) by using any DP or other algorithm
   Sample initial state $s_0^l \sim \nu$
   **for** $h = 0, \cdots, H-1$ **do**
      Take action $a_h^t \sim \tilde{\pi}^t(|s_h^t)$
      Observe $(s_{h+1}^t, r_h^t)$
   **end for**
**end for**

---

$\tilde{Q}(\cdot, \cdot; \theta^*) = Q(\cdot, \cdot; \theta^*)$, and $\beta(s) := \beta \geq 0$ (i.e., it is state-invariant) is also known to the expert. Thus, the learning agent's posterior distribution over $\theta^*$ given the offline dataset is,

$$\mu_1(\theta^* \in \cdot) := \mathbb{P}(\theta^* \in \cdot|\mathcal{D}_0) \propto \mathbb{P}(\mathcal{D}_0|\theta^* \in \cdot)\mathbb{P}(\theta^* \in \cdot)$$

$$= \mathbb{P}(\theta^* \in \cdot) \times \int_{\theta \in \cdot} \prod_l^L \prod_{h=0}^{H-1} \theta(\bar{s}_{h+1}^l|\bar{s}_h^l, \bar{a}_h^l)\pi_h^\beta(\bar{a}_h^l|\bar{s}_h^l, \theta)\nu(\bar{s}_0^l)\, d\theta. \tag{3}$$

A PSRL agent (Osband et al., 2013; Ouyang et al., 2017) takes this as the prior, and then updates the posterior distribution over $\theta^*$ as online observation tuples, $\{(s_h^t, a_h^t, s_h^{t'}, r_h^t)_{h=0}^{H-1}$ become available. Such an agent is really an ideal agent with assumed posterior distribution updates being exact. In practice, this is computationally intractable and we will need to get samples from an approximate posterior distribution, an issue which we will address in the next section. We will denote the posterior distribution over $\theta^*$ given the online observations by episodes $t$ and the offline dataset by $\mu_t$.

We note the key steps (A1)-(A2) in the algorithm above where we use both offline dataset $\mathcal{D}_0$ and online observations $\mathcal{H}_t$ to compute the posterior distribution over $\theta^*$ by use of the prior $\nu$.

### 3.1 Prior-dependent Regret Bound

It is natural to expect some regret reduction if an offline demonstration dataset is available to warm-start the online learning. However, the degree of improvement must depend on the "quality" of demonstrations, for example through the competence parameter $\beta$. Further note that the role of the offline dataset is via the prior distribution the PSRL algorithm uses. Thus, theoretical analysis involves obtaining a prior-dependent regret bound, which we obtain next.

**Lemma 3.1.** *Let $\varepsilon = \mathbb{P}(\tilde{\pi}^1 \neq \pi^*)$, i.e. the probability that the strategy used for the first episode, $\tilde{\pi}^1$ is not optimal. Then,*

$$\mathfrak{BR}_T(\phi^{iPSRL}) = O(\sqrt{\varepsilon H^4 S^2 AT}(1 + \log(T))). \tag{4}$$

The proof can be found in the Appendix. Note that this Lemma provides a prior dependent-bound in terms of $\varepsilon$.

In the rest of the section, we provide an upper bound on $\varepsilon = \mathbb{P}(\tilde{\pi}^1 \neq \pi^*)$ for **Algorithm 1: iPSRL.** We first show that $\varepsilon$ can be bounded in terms of estimation error of the optimal strategy given the offline data.

**Lemma 3.2.** *Let $\hat{\pi}^*$ be any estimator of $\pi^*$ constructed from $\mathcal{D}_0$, then $\mathbb{P}(\tilde{\pi}^1 \neq \pi^*) \leq 2\mathbb{P}(\hat{\pi}^* \neq \pi^*)$.*

*Proof.* If $\tilde{\pi}^1 \neq \pi^*$, then either $\hat{\pi}^* \neq \tilde{\pi}^1$ or $\hat{\pi}^* \neq \pi^*$ must be true. Conditioning on $\mathcal{D}_0$, $\hat{\pi}^*$ is identically distributed as $\pi^*$ while $\hat{\pi}^*$ is deterministic, therefore

$$\mathbb{P}(\tilde{\pi}^1 \neq \pi^*) \leq \mathbb{P}(\hat{\pi}^* \neq \tilde{\pi}^1) + \mathbb{P}(\hat{\pi}^* \neq \pi^*) = 2\mathbb{P}(\hat{\pi}^* \neq \pi^*)$$

$\square$

We now bound the estimation error of $\pi^*$. We assume the following about the prior distribution of $\theta^*$.

**Assumption 3.3.** There exists a $\Delta > 0$ such that for all $\theta \in \Theta$, $h = 0, 1, \cdots, H-1$, and $s \in \mathcal{S}$, there exists an $a^* \in \mathcal{A}$ such that $Q_h(s, a^*; \theta) \geq Q_h(s, a'; \theta) + \Delta, \ \forall a' \in \mathcal{A} \backslash \{a^*\}$.

Define $p_h(s; \theta) := \mathbb{P}_{\theta, \pi^*(\theta)}(s_h = s)$, where $\pi^*(\theta)$ denotes the optimal policy under model $\theta$.

**Assumption 3.4.** The infimum probability of any reachable non-final state , defined as

$$\underline{p} := \inf\{p_h(s; \theta) : 0 \leq h < H, s \in \mathcal{S}, \theta \in \Theta, p_h(s; \theta) > 0\}$$

satisfies $\underline{p} > 0$.

We now describe a procedure to construct an estimator of the optimal policy, $\hat{\pi}^*$ from $\mathcal{D}_0$ so that $\mathbb{P}(\pi^* \neq \hat{\pi}^*)$ is small. Fix an integer $L$, and choose a $\delta \in (0, 1)$. For each $\theta \in \Theta$, define a deterministic Markov policy $\pi^*(\theta) = (\pi_h^*(\cdot; \theta))_{h=0}^{H-1}$ sequentially through

$$\pi_h^*(s; \theta) = \begin{cases} \arg\max_a Q_h(s, a; \theta), & \text{if } \mathbb{P}_{\theta, \pi_{0:h-1}^*(\theta)}(s_h = s) > 0 \\ \bar{a}_0, & \text{if } \mathbb{P}_{\theta, \pi_{0:h-1}^*(\theta)}(s_h = s) = 0, \end{cases} \tag{5}$$

where the tiebreaker for the argmax operation is based on a fixed order on actions, and $\bar{a}_0 \in \mathcal{A}$ is a fixed action in $\mathcal{A}$. It is clear that $\pi^*(\theta)$ is an optimal policy for the MDP $\theta$. Furthermore, for those states that are impossible to be visited, we choose to take a fixed action $\bar{a}_0$. Although the choice of action at those states doesn't matter, our construction will be helpful for the proofs.

**Construction of $\hat{\pi}^*$:** Let $N_h(s)$ (resp. $N_h(s, a)$) be the number of times state $s$ (resp. state-action pair $(s, a)$) appears at time $h$ in dataset $\mathcal{D}_0$. Define $\hat{\pi}^*$ to be such that:

- $\hat{\pi}_h^*(s) = \arg\max_{a \in \mathcal{A}} N_h(s, a)$ (ties are broken through some fixed ordering of actions) whenever $N_h(s) \geq \delta L$;

- $\hat{\pi}_h^*(s) = \bar{a}_0$ whenever $N_h(s) < \delta L$. $\bar{a}_0$ is a fixed action in $\mathcal{A}$ that was used in the definition of $\pi^*(\theta)$.

The idea of the proof is that for sufficiently large $\beta$ and $L$, we can choose a $\delta \in (0, 1)$ such that

- *Claim 1:* If $s \in \mathcal{S}$ is probable at time $h$ under $\pi^*(\theta)$, then $N_h(s) \geq \delta L$ with large probability. Furthermore, $\pi_h^*(s) = \arg\max_{a \in \mathcal{A}} N_h(s, a)$ with large probability as well.

- *Claim 2:* If $s \in \mathcal{S}$ is improbable at time $h$ under $\pi^*(\theta)$, then $N_h(s) < \delta L$ with large probability;

Given the two claims, we can then conclude that the probability of $\pi^* \neq \hat{\pi}^*$ is small via a standard union bound argument. The arguments are formalized in Lemma A.2. We now present the upper bound on Bayesian regret for **Algorithm 1: iPSRL**.

**Theorem 3.5.** *For sufficiently large $\beta$ independent of $L$, we have*

$$\mathfrak{BR}_T(\phi^{iPSRL}) = O(\sqrt{\varepsilon_L H^4 S^2 A T}(1 + \log(T))). \tag{6}$$

*where*

$$\varepsilon_L = \min\left\{1, 2SH\left[\exp\left(-\frac{L\underline{p}^2}{18}\right) + \exp\left(-\frac{L\underline{p}}{36}\right)\right]\right\}.$$

*Proof.* The result follows from Lemma 3.1, Lemma 3.2, and Lemma A.2. $\square$

We note that the right-hand side of Eq. (6) converges to zero exponentially fast as $N = LH \to \infty$.

*Remark* 3.6. (a) For fixed $N$, and large $S$ and $A$, the regret bound is $\tilde{O}(H^2 S \sqrt{AT})$, which possibly could be improved in $H$. (b) For a suitably large $\beta$, the regret bound obtained goes to zero exponentially fast as $L$, the number of episodes in the offline dataset, goes to infinity thus indicating the online learning algorithm's ability to learn via imitation of the expert.

# 4 Approximating iPSRL

## 4.1 The Informed RLSVI Algorithm

The `iPSRL` algorithm introduced in the previous section assumes that posterior updates can be done exactly. In practice, the posterior update in Eq. (3) is challenging due to the loss of conjugacy while using the Bayes rule. Thus, we must find a computationally efficient way to do approximate posterior updates (and obtain samples from it) to enable practical implementation. Hence, we propose a novel approach based on Bayesian bootstrapping to obtain approximate posterior samples. The key idea is to perturb the loss function for the maximum a posterior (MAP) estimate and use the point estimate as a surrogate for the exact posterior sample.

Note that in the ensuing, we regard $\beta$ as also unknown to the learning agent (and $\lambda = \infty$ for simplicity). Thus, the learning agent must form a belief over both $\theta$ and $\beta$ via a joint posterior distribution conditioned on the offline dataset $\mathcal{D}_0$ and the online data at time $t$, $\mathcal{H}_t$. We denote the prior pdf over $\theta$ by $f(\cdot)$ and prior pdf over $\beta$ by $f_2(\cdot)$.

For the sake of compact notation, we denote $Q_h^*(s, a; \theta)$ as $Q_h^\theta(s, a)$ in this section. Now, consider the offline dataset,

$$\mathcal{D}_0 = \{((s_h^l, a_h^l, \check{s}_h^l, r_h^l)_{h=0}^{H-1})_{l=1}^L\}$$

and denote $\theta = (\theta_h)_{h=0}^{H-1}$. We introduce the *temporal difference error* $\mathcal{E}_h^l$ (parameterized by a given $Q^\theta$),

$$\mathcal{E}_h^l(Q^\theta) := \left( r_h^l + \max_b Q_{h+1}^\theta(\check{s}_h^l, b) - Q_h^\theta(s_h^l, a_h^l) \right).$$

We will regard $Q_h^\theta$ to only be parameterized by $\theta_h$, i.e., $Q_h^{\theta_h}$ but abuse notation for the sake of simplicity. We use this to construct a *parameterized offline dataset*,

$$\mathcal{D}_0(Q^\theta) = \{((s_h^l, a_h^l, \check{s}_h^l, \mathcal{E}_h^l(Q^\theta))_{h=0:H-1})_{l=1:L}\}.$$

A parametrized online dataset $\mathcal{H}_t(Q^\theta)$ after episode $t$ can be similarly defined. To ease notation, we will regard the $j$th episode during the online phase as the $(L+j)$th observed episode. Thus,

$$\mathcal{H}_t(Q^\theta) = \{((s_h^k, a_h^k, \check{s}_h^k, \mathcal{E}_h^k(Q^\theta))_{h=0:H-1})_{k=L+1:L+t}\},$$

the dataset observed during the online phase by episode $t$.

Note that $Q^\theta$ is to be regarded as a parameter. Now, at time $t$, we would like to obtain a **MAP estimate** for $(\theta, \beta)$ by solving the following:

$$\textbf{MAP:} \quad \arg\max_{\theta, \beta} \log P(\mathcal{H}_t(Q^\theta)|\mathcal{D}_0(Q^\theta), \theta, \beta) + \log P(\mathcal{D}_0(Q^\theta)|\theta, \beta) + \log f(\theta) + \log f_2(\beta). \tag{7}$$

Denote a perturbed version of the $Q^\theta$-parameterized offline dataset by

$$\tilde{\mathcal{D}}_0(Q^\theta) = \{((s_h^l, \tilde{a}_h^l, \check{s}_h^l, \tilde{\mathcal{E}}_h^l)_{h=0:H-1})_{l=1:L}\}$$

where random perturbations are added: (i) actions have perturbation $w_l^h \sim \exp(1)$, (ii) rewards have perturbations $z_h^l \sim \mathcal{N}(0, \sigma^2)$, and (iii) the prior $\tilde{\theta} \sim \mathcal{N}(0, \Sigma_0)$.

Note that the first and second terms involving $\mathcal{H}_t$ and $\mathcal{D}_0$ in Eq. (7) are independent of $\beta$ when conditioned on the actions. Thus, we have a sum of *log-likelihood of TD error, transition and action* as follows:

$$\log P(\tilde{\mathcal{D}}_0(Q^\theta)|Q_{0:H}^\theta) = \sum_{l=1}^L \sum_h \left( \log P(\tilde{\mathcal{E}}_h^l|\check{s}_h^l, a_h^l, s_h^l, Q_{0:H}^\theta) + \log P(\check{s}_h^l|a_h^l, s_h^l, Q_{0:H}^\theta) + \log P(a_h^l|s_h^l, Q_{0:H}^\theta) \right)$$

$$\leq \sum_{l=1}^L \sum_h \left( \log P(\tilde{\mathcal{E}}_h^l|\check{s}_h^l, a_h^l, s_h^l, Q_{h:h+1}^\theta) + \log \pi_h^\beta(a_h^l|s_h^l, Q_h^\theta) \right).$$

By ignoring the log-likelihood of the transition term (akin to optimizing an upper bound on the negative loss function), we are actually being *optimistic*.

For the terms in the upper bound above, under the random perturbations assumed above, we have

$$\log P(\tilde{\mathcal{E}}_h^l | \check{s}_h^l, a_h^l, s_h^l, Q_{h:h+1}^\theta) = -\frac{1}{2}\left(r_h^l + z_h^l + \max_b Q_{h+1}^\theta(\check{s}_h^l, b) - Q_h^\theta(s_h^l, a_h^l)\right)^2 + \text{ constant}$$

and

$$\log \pi_h^\beta(a_h^l | s_h^l, Q_h^\theta) = w_h^l\left(\beta Q_h^\theta(s_h^l, a_h^l) - \log\sum_b \exp\left(\beta Q_h^\theta(s_h^l, b)\right)\right).$$

Now, denote a perturbed version of the $Q^\theta$-parametrized online dataset,

$$\tilde{\mathcal{H}}_t(Q^\theta) = \{((s_h^k, a_h^k, \check{s}_h^k, \tilde{\mathcal{E}}_h^k)_{h=0:H-1})_{k=L+1:L+t}\},$$

and thus similar to before, we have

$$\log P(\tilde{\mathcal{H}}_t(Q^\theta) | \tilde{\mathcal{D}}_0(Q^\theta), Q_{0:H}^\theta) = \sum_{k=L+1}^{L+t}\sum_h \left(\log P(\tilde{\mathcal{E}}_h^k(Q^\theta) | \check{s}_h^k, a_h^k, s_h^k, Q_{0:H}^\theta) + \log P(\check{s}_h^k | a_h^k, s_h^k, Q^\theta)\right),$$

$$\leq \sum_{k=L+1}^{L+t}\sum_h \left(\log P(\tilde{\mathcal{E}}_h^k | \check{s}_h^k, a_h^k, s_h^k, Q_{h:h+1}^\theta)\right),$$

where we again ignored the transition term to obtain an *optimistic* upper bound.

Given the random perturbations above, we have

$$\log P(\tilde{\mathcal{E}}_h^k(Q^\theta) | \check{s}_h^k, a_h^k, s_h^k, Q_{h:h+1}^\theta) = -\frac{1}{2}\left(r_h^k + z_h^k + \max_b Q_{h+1}^\theta(\check{s}_h^k, b) - Q_h^\theta(s_h^k, a_h^k)\right)^2 + \text{ constant}.$$

The prior over $\beta$, $f_2(\beta)$ is assumed to be an exponential pdf $\lambda_2 \exp(-\lambda_2\beta), \beta \geq 0$, while that over $\theta$ is assumed Gaussian. Thus, putting it all together, we get the following ***optimistic loss function*** (to minimize over $\theta$ and $\beta$),

$$\begin{aligned}
\tilde{\mathcal{L}}(\theta, \beta) = &\frac{1}{2\sigma^2}\sum_{k=1}^{L+t}\sum_{h=0}^{H-1}\left(r_h^k + z_h^k + \max_b Q_{h+1}^\theta(\check{s}_h^k, b) - Q_h^\theta(s_h^k, a_h^k)\right)^2 \\
&- \sum_{l=1}^{L}\sum_{h=0}^{H-1} w_h^l\left(\beta Q_h^\theta(s_h^l, a_h^l) - \log\sum_b \exp\left(\beta Q_h^\theta(s_h^l, b)\right)\right) + \frac{1}{2}(\theta - \tilde{\theta})^\top\Sigma_0(\theta - \tilde{\theta}) + \lambda_2\beta.
\end{aligned} \tag{8}$$

The above loss function is difficult to optimize in general due to the max operation, and the $Q$-value function in general having a nonlinear form.

Now **Algorithm 2: iRLSVI** can be summarized by replacing steps (A1)-(A2) in **Algorithm 1: iPSRL** by (B1)-(B2), with the other steps being the same:

> (B1) Solve the MAP Problem (7) by minimizing loss function (8).
> (B2) Get solutions $(\tilde{\theta}_l, \tilde{\beta}_l)$.

*Remark* 4.1. Note that the loss function in Eq. (8) can be hard to jointly optimize over $\theta$ and $\beta$. In particular, estimates of $\beta$ can be quite noisy when $\beta$ is large, and the near-optimal expert policy only covers the state-action space partially. Thus, we consider other methods of estimating $\beta$ that are more robust, which can then be plugged into the loss function in Eq. (8). Specifically, we could simply look at the entropy of the empirical

distribution of the action in the offline dataset. Suppose the empirical distribution of $\{\bar{a}_0^l, \ldots \bar{a}_H^l\}_{l=1}^L$ is $\mu_A$. Then we use $c_0/\mathcal{H}(\mu_A)$ as an estimation for $\beta$, where $c_0 > 0$ is a hyperparameter. The intuition is that for smaller $\beta$, the offline actions tend to be more uniform and thus the entropy will be large. This is an unsupervised approach and agnostic to specific offline data generation process.

*Remark* 4.2. In the loss function in Eq. (8), the parameter $\theta$ appears inside the max operation. Thus, it can be quite difficult to optimize over $\beta$. Since the loss function is typically optimized via an iterative algorithm such as a gradient descent method, a simple and scalable solution that works well in practice is to use the parameter estimate $\theta$ from the previous iteration inside the max operation, and thus optimize over $\theta$ only in the other terms.

## 4.2 iRLSVI bridges Online RL and Imitation Learning

In the previous subsection, we derived `iRLSVI`, a Bayesian-bootstrapped algorithm. We now present interpretation of the algorithm as bridging online RL (via commonality with the RLSVI algorithm (Osband et al., 2016a) and imitation learning, and hence a way for its generalization.

Consider the RLSVI algorithm for online reinforcement learning as introduced in (Osband et al., 2019). It draws its inspiration from the posterior sampling principle for online learning, and has excellent cumulative regret performance. RLSVI, that uses all of the data available at the end of episode $t$, including any offline dataset involves minimizing the corresponding loss function at each time step:

$$\tilde{\mathcal{L}}_{\mathrm{RLSVI}}(\theta) = \frac{1}{2\sigma^2} \sum_{k=1}^{L+t} \sum_{h=0}^{H-1} \left( r_h^k + \max_b Q_{h+1}^\theta(\check{s}_h^k, b) - Q_h^\theta(s_h^k, a_h^k) \right)^2 + \frac{1}{2}(\theta_{0:H} - \tilde{\theta}_{0:H})^\top \Sigma_0 (\theta_{0:H} - \tilde{\theta}_{0:H}).$$

Now, let us consider an imitation learning setting. Let $\tau_l = (s_h^l, a_h^l, \check{s}_h^l)_{h=0}^{H-1}$ be the trajectory of the $l$th episode. Let $\hat{\pi}_h(a|s)$ denote the empirical estimate of probability of taking action $a$ in state $s$ at time $h$, i.e., an empirical estimate of the expert's randomized policy. Let $p(\tau)$ denote the probability of observing the trajectory under the policy $\hat{\pi}$.

Let $\pi_h^{\beta,\theta}(\cdot|s)$ denote the parametric representation of the policy used by the expert. And let $p^{\beta,\theta}(\tau)$ denote the probability of observing the trajectory $\tau$ under the policy $\pi^{\beta,\theta}$. Then, the loss function corresponding to the KL divergence between $\Pi_{l=1}^L p(\tau_l)$ and $\Pi_{l=1}^L p^{\beta,\theta}(\tau_l)$ is given by

$$\tilde{\mathcal{L}}_{\mathrm{IL}}(\beta, \theta) = D_{KL}\left( \Pi_{l=1}^L p(\tau_l) || \Pi_{l=1}^L p^{\beta,\theta}(\tau_l) \right) = \int \Pi_{l=1}^L p(\tau_l) \log \frac{\Pi_{l=1}^L p(\tau_l)}{\Pi_{l=1}^L p^\beta(\tau_l)} = \sum_{l=1}^L \int p(\tau_l) \log \frac{p(\tau_l)}{p^{\beta,\theta}(\tau_l)},$$

$$= \sum_{l=1}^L \sum_{h=0}^{H-1} \log \frac{\hat{\pi}_h(a_h^l|s_h^l)}{\pi_h^{\beta,\theta}(a_h^l|s_h^l)}$$

$$= \sum_{l=1}^L \sum_{h=0}^{H-1} [\log \hat{\pi}_h(a_h^l|s_h^l) - \log \pi_h^{\beta,\theta}(a_h^l|s_h^l)]$$

$$= -\sum_{l=1}^L \sum_{h=0}^{H-1} \left( \beta Q_h^\theta(s_h^l, a_h^l) - \log \sum_b \exp\left( \beta Q_h^\theta(s_h^l, b) \right) \right) + \text{constant}.$$

*Remark* 4.3. (i) The loss function $\tilde{\mathcal{L}}_{\mathrm{IL}}(\beta, \theta)$ is the same as the second (action-likelihood) term in Eq. (8) while the loss function $\tilde{\mathcal{L}}_{\mathrm{RLSVI}}(\theta)$ is the same as the first and third terms there (except for perturbation) and minus the $\lambda_2 \beta$ term that corresponds to the prior over $\beta$. (ii) Note that while we used the more common KL divergence for the imitation learning loss function, use of log loss would yield the same outcome.

Thus, the `iRLSVI` loss function can be viewed as

$$\tilde{\mathcal{L}}(\beta, \theta) = \tilde{\mathcal{L}}_{\mathrm{RLSVI}}(\theta) + \tilde{\mathcal{L}}_{\mathrm{IL}}(\beta, \theta) + \lambda_2 \beta, \tag{9}$$

thus establishing that the proposed algorithm may be viewed as bridging Online RL with Imitation Learning. Note that the last term corresponds to the prior over $\beta$. If $\beta$ is known (or uniform), it will not show up in the loss function above.

The above also suggests a possible way to generalize and obtain other online learning algorithms that can bootstrap by use of offline datasets. Namely, at each step, they can optimize a general loss function of the following kind:

$$\tilde{\mathcal{L}}_\alpha(\beta, \theta) = \alpha \tilde{\mathcal{L}}_{\mathrm{ORL}}(\theta) + (1 - \alpha) \tilde{\mathcal{L}}_{\mathrm{IL}}(\beta, \theta) + \lambda_2 \beta, \tag{10}$$

where $\tilde{\mathcal{L}}_{\mathrm{ORL}}$ is a loss function for an Online RL algorithm, $\tilde{\mathcal{L}}_{\mathrm{IL}}$ is a loss function for some Imitation Learning algorithm, and factor $\alpha \in [0, 1]$ provides a way to tune between emphasizing the offline imitation learning and the online reinforcement learning.

## 5 Empirical Results

**Experimental setup.** We now present some empirical results in two prototypical environments: "Deep Sea" and "Maze". Specifically, we compare three variants of the RLSVI agents, which are respectively referred to as *informed* RLSVI (iRLSVI), *partially informed* RLSVI (piRLSVI), and *uninformed* RLSVI (uRLSVI). All three agents are tabular RLSVI agents with similar posterior sampling-type exploration schemes. However, they differ in whether or not and how to exploit the offline dataset. In particular, uRLSVI ignores the offline dataset; piRLSVI exploits the offline dataset but does not utilize the information about the generative policy; while iRLSVI fully exploits the information in the offline dataset, about both the generative policy and the reward feedback. Please refer to Appendix D.1 for the pseudo-codes of these agents. We note no other algorithms are known for the problem as posed.

**Deep Sea** (Figure 1 (i))(Osband et al., 2019) is an episodic reinforcement learning problem with state space $\mathcal{S} = \{0, 1, \ldots, M\}^2$ and , where $M$ is its size. The state at period $h$ in episode $t$ is $s_h^t = (x_h^t, d_h^t) \in \mathcal{S}$, where $x_h^t = 0, 1, \ldots, M$ is the horizontal position while $d_h^t = 0, 1, \ldots, M$ is the depth (vertical position). Its action space is $\mathcal{A} = \{\texttt{left}, \texttt{right}\}$ and time horizon length is $H = M$. Its reward function is as follows: If the agent chooses an action $\texttt{right}$ in period $h < H$, then it will receive a reward $-0.1/M$, which corresponds to a "small cost"; If the agent successfully arrives at state $(M, M)$ in period $H = M$, then it will receive a reward 1, which corresponds to a "big bonus"; otherwise, the agent will receive reward 0. The system dynamics are as follows: for period $h < H$, the agent's depth in the next period is always increased by 1, i.e., $d_{h+1}^t = d_h^t + 1$. For the agent's horizontal position, if $a_h^t = \texttt{left}$, then $x_{h+1}^t = \max\{x_h^t - 1, 0\}$, i.e., the agent will move left if possible. On the other hand, if $a_h^t = \texttt{right}$, then we have $x_{h+1}^t = \min\{x_h^t + 1, M\}$ with prob. $1 - 1/M$ and $x_{h+1}^t = x_h^t$ with prob. $1/M$. The initial state of this environment is fixed at state $(0, 0)$. In this section, we fix the size of Deep Sea as $M = 10$.

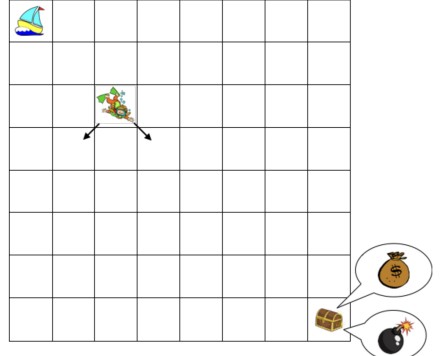 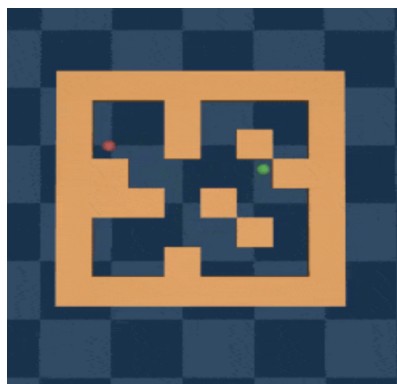

Figure 1: (i) DeepSea Environment with start state in top left corner, and goal state in bottom right corner. (ii) The map for the Maze environment.

**Maze** (Figure 1 (ii)) is also an episodic reinforcement learning problem, which is a variant of a maze problem proposed in D4RL (Fu et al., 2020). To fit this problem into the finite-state framework of this paper, we discretize the locations in Figure 1 (ii) into a $6 \times 6$ grid. Note that excluding the walls, there are 26 valid locations in the maze. We set the time horizon of this problem as $H = 7$. Since each state is a location-period

pair, there are $26 \times 7 = 182$ states in this problem. Moreover, we assume that this problem has a fixed initial state: in each episode, the agent starts at the location denoted by the green dot in Figure 1 (ii).

We assume that the action space for Maze is $\mathcal{A} = \{\texttt{left}, \texttt{right}, \texttt{up}, \texttt{down}, \texttt{stay}\}$. At each period $h < H - 1$, if the agent takes action $\texttt{left}$, and it is also feasible to go left (i.e. the agent does not hit the wall if going left), then the agent will successfully go left with probability 0.95, and fail to go left and stay at the same location with probability 0.05. On the other hand, if it is infeasible to go left, then the agent will stay at the same location with probability 1. The state transitions under action $\texttt{right}$, $\texttt{up}$, and $\texttt{down}$ are defined similarly, with the same success/failure probabilities. Finally, if the agent takes action $\texttt{stay}$, then it will stay at the same location with probability 1. The reward function of Maze is as follows: if the agent is at the target location, which is denoted by the red dot in Figure 1 (ii), then the agent will receive a "big bonus" of reward 1; otherwise, the agent will incur a "small cost", which is reward $-0.01$.

In both environments, the offline dataset is generated based on the expert's policy specified in Eq. (2), and we assume $\beta(s) = \beta$ (a constant) across all states. We set the size of the offline dataset $\mathcal{D}_0$ as $|\mathcal{D}_0| = \kappa |\mathcal{A}||\mathcal{S}|$, where $\kappa \geq 0$ is referred to as *data ratio*.

**Experimental results.** We run the experiments on Deep Sea for $T = 300$ episodes, and run the experiments on Maze for $T = 200$ episodes. For both environments, the empirical cumulative regrets are averaged over 50 simulations. The experimental results are illustrated in Figure 2 and 3, as well as Figure 5 in Appendix D.2 and Figure 6 in Appendix D.3.

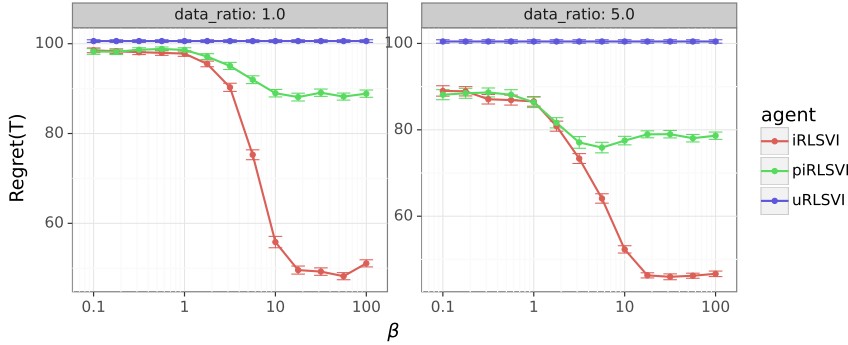

Figure 2: Cumulative regret vs. $\beta$ in Deep Sea.

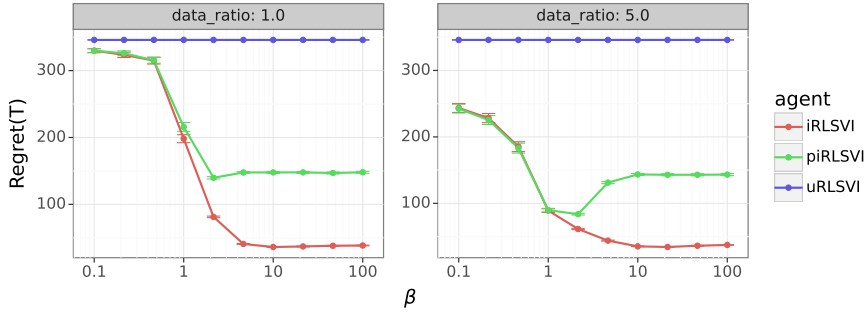

Figure 3: Cumulative regret vs. $\beta$ in Maze.

Specifically, Figure 2 plots the cumulative regret on Deep Sea in the first $T = 300$ episodes as a function of the expert's deliberateness $\beta$, for two different data ratios, $\kappa = 1$, and 5. There are several interesting observations based on Figure 2: (i) Figure 2 shows that $\texttt{iRLSVI}$ and $\texttt{piRLSVI}$ tend to perform much better than $\texttt{uRLSVI}$, which demonstrates the advantages of exploiting the offline dataset, and this improvement tends to be more dramatic with a larger offline dataset. (ii) When we compare $\texttt{iRLSVI}$ and $\texttt{piRLSVI}$, we note

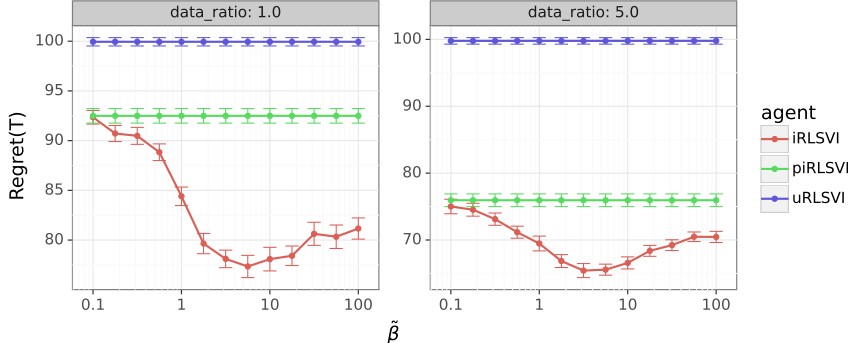

Figure 4: Robustness of `iRLSVI` to misspecification.

that their performance is similar when $\beta$ is small, but `iRLSVI` performs much better than `piRLSVI` when $\beta$ is large. This is because when $\beta$ is small, the expert's generative policy does not contain much information; and as $\beta$ gets larger, it contains more information and eventually it behaves like imitation learning and learns the optimal policy as $\beta \to \infty$. Note that the error bars denote the standard errors of the empirical cumulative regrets, hence the improvements are statistically significant.

Similarly, Figure 3 plots the cumulative regret on Maze in the first $T = 200$ episodes as a function of $\beta$, also for data ratios $\kappa = 1$ and 5. Note that we have obtained similar results in this experiment, and the improvements are also statistically significant.

**Robustness to misspecification of $\beta$.** We also investigate the robustness of various RLSVI agents with respect to the possible misspecification of $\beta$. In particular, we demonstrate empirically that in the Deep Sea environment with $M = 10$, with offline dataset is generated by an expert with deliberateness $\beta = 5$, the `iRLSVI` agent is quite robust to moderate misspecification. Here, the misspecified deliberateness parameter is denoted $\tilde{\beta}$. The empirical results are illustrated in Figure 4, where the experiment is run for $T = 300$ episodes and the empirical cumulative regrets are averaged over 50 simulations.

Since `uRLSVI` and `piRLSVI` do not use parameter $\tilde{\beta}$, thus, as expected, their performance is constant over $\tilde{\beta}$. On the other hand, `iRLSVI` explicitly uses parameter $\tilde{\beta}$. As Figure 4 shows, the performance of `iRLSVI` does not vary much as long as $\tilde{\beta}$ has the same order of magnitude as $\beta$. However, there will be significant performance loss when $\tilde{\beta}$ is too small, especially when the data ratio is also small. This makes sense since when $\tilde{\beta}$ is too small, `iRLSVI` will choose to ignore all the information about the generative policy and eventually reduces to `piRLSVI`. Similar results can be anticipated for the Maze environment and are omitted to save space.

## 6 Conclusions

In this paper, we have introduced and studied the following problem: Given an offline demonstration dataset from an imperfect expert, what is the best way to leverage it to bootstrap online learning performance in MDPs. We have followed a principled approach and introduced two algorithms: the ideal `iPSRL` algorithm, and the `iRLSVI` algorithm that is computationally practical and seamlessly bridges online RL and imitation learning in a very natural way. We have shown significant reduction in regret both empirically, and theoretically as compared to two natural baselines. The dependence of the regret bound on some of the parameters (e.g., $H$) could be improved upon, and is a good direction for future work. In future work, we will also combine the `iRLSVI` algorithm with deep learning to leverage offline datasets effectively for continuous state and action spaces as well.

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

## A  Auxiliary Lemmas

**Lemma A.1.** *Let $X$ be the sum of $L$ i.i.d. Bernoulli random variables with mean $p \in (0,1)$. Let $q \in (0,1)$, then*

$$\mathbb{P}(X \leq qL) \leq \exp\left(-2L(q-p)^2\right), \qquad \text{if } q < p,$$
$$\mathbb{P}(X \geq qL) \leq \exp\left(-2L(q-p)^2\right), \qquad \text{if } q > p.$$

*Proof.* Both inequalities can be obtained by applying Hoeffding's Inequality. $\square$

**Lemma A.2.** *Let $\Delta$ and $\underline{p}$ be as in Assumptions 3.3 and 3.4 respectively and let*

$$\underline{\beta} := [\log 3 - \log \underline{p} + \log(H-1) + \log(A-1)]/\Delta.$$

*For any $\beta \geq \underline{\beta}$ and $N \in \mathbb{N}$, there exists an estimator $\hat{\pi}^*$ constructed from $\mathcal{D}_0$ that satisfies*

$$\mathbb{P}(\pi^* \neq \hat{\pi}^*) \leq SH\left[\exp\left(-\frac{L\underline{p}^2}{18}\right) + \exp\left(-\frac{L\underline{p}}{36}\right)\right].$$

The proof is available in Appendix C.

## B  Proof of Lemma 3.1

Let $\tilde{\theta}^k$ the environment sampled for the $k$th episode, and $\tilde{\pi}^k$ be the optimal strategy under $\tilde{\theta}^k$. Recall that $\mathcal{H}_{k-1}$ is the data available to the iPSRL agent before the start of learning episode $k$. For notational simplicity, let $\mathbb{P}_k(\cdot) = \mathbb{P}(\cdot|\mathcal{H}_{k-1})$ and $\mathbb{E}_k[\cdot] = \mathbb{E}[\cdot|\mathcal{H}_{k-1}]$. We first prove the following Lemma.

**Lemma B.1.** $\mathbb{P}(\tilde{\pi}^k \neq \pi^*) \leq \mathbb{P}(\tilde{\pi}^1 \neq \pi^*)$ *for all $k \geq 1$.*

*Proof.* Define $f(x) = x(1-x)$. $f$ is a concave function. We have

$$\mathbb{P}(\tilde{\pi}^k \neq \pi^*) = \mathbb{E}[\mathbb{P}_k[\tilde{\pi}^k \neq \pi^*]] = \mathbb{E}\left[\sum_{\pi \in \Pi} \mathbb{P}_k(\tilde{\pi}^k = \pi, \pi^* \neq \pi)\right] \tag{11}$$

$$= \mathbb{E}\left[\sum_{\pi \in \Pi} f(\mathbb{P}_k(\pi^* = \pi))\right] \tag{12}$$

$$= \mathbb{E}\left[\sum_{\pi \in \Pi} \mathbb{E}_1[f(\mathbb{P}_k(\pi^* = \pi))]\right] \leq \mathbb{E}\left[\sum_{\pi \in \Pi} f(\mathbb{E}_1[\mathbb{P}_k(\pi^* = \pi)])\right] \tag{13}$$

$$= \mathbb{E}\left[\sum_{\pi \in \Pi} f(\mathbb{P}_1(\pi^* = \pi))\right] = \mathbb{E}\left[\sum_{\pi \in \Pi} \mathbb{P}_1(\tilde{\pi}^1 = \pi, \pi^* \neq \pi)\right] \tag{14}$$

$$= \mathbb{P}(\tilde{\pi}^1 \neq \pi^*) \tag{15}$$

$\square$

Now we proceed to prove Lemma 3.1.

Let $J_\pi^\theta := \mathbb{E}_{\theta,\pi}[\sum_{h=0}^{H-1} r_h(s_h, a_h) + r_H(s_H)]$ denote the expected total reward under the environment $\theta$ and the Markov strategy $\pi$. Define

$$Z_k := J_{\pi^*}^{\theta^*} - J_{\tilde{\pi}^k}^{\theta^*} \tag{16}$$

$$\tilde{Z}_k := J_{\tilde{\pi}^k}^{\tilde{\theta}^k} - J_{\tilde{\pi}^k}^{\theta^*} \tag{17}$$

$$I_k := \mathbf{1}_{\{\tilde{\pi}^k \neq \pi^*\}} \tag{18}$$

First, note that $Z_k = Z_k I_k$ with probability 1, hence

$$\mathbb{E}_k[Z_k - \tilde{Z}_k I_k] = \mathbb{E}_k[(Z_k - \tilde{Z}_k)I_k] = \mathbb{E}_k[(J^{\theta^*}_{\pi^*} - J^{\tilde{\theta}^k}_{\tilde{\pi}^k})I_k] \tag{19}$$

$$= \mathbb{E}_k[J^{\theta^*}_{\pi^*}\mathbf{1}_{\{\tilde{\pi}^k \neq \pi^*\}}] - \mathbb{E}_k[J^{\tilde{\theta}^k}_{\tilde{\pi}^k}\mathbf{1}_{\{\pi^* \neq \tilde{\pi}^k\}}] = 0, \tag{20}$$

where the last equality is true since $\theta^*$ and $\tilde{\theta}^k$ are independently identically distributed given $\mathcal{D}_k$. Therefore, we can write the Bayesian regret as $\mathbb{E}[\sum_{k=1}^{T} \tilde{Z}_k I_k]$. By Cauchy-Schwartz inequality, we have

$$\mathbb{E}\left[\sum_{k=1}^{T} \tilde{Z}_k I_k\right] \leq \sqrt{\left(\sum_{k=1}^{T} \mathbb{E}[I_k^2]\right)\left(\sum_{k=1}^{T} \mathbb{E}[\tilde{Z}_k^2]\right)} \tag{21}$$

Using Lemma B.1, the first part can be bounded by

$$\sum_{k=1}^{T} \mathbb{E}[I_k^2] = \sum_{k=1}^{T} \mathbb{P}(\tilde{\pi}^k \neq \pi^*) \leq T\mathbb{P}(\tilde{\pi}^1 \neq \pi^*) = \varepsilon T \tag{22}$$

The rest of the proof provides a bound on $\sum_{k=1}^{T} \mathbb{E}[\tilde{Z}_k^2]$.

Let $\mathcal{T}^{\theta}_{\pi_h}$ be the Bellman operator at time $h$ defined by $\mathcal{T}^{\theta}_{\pi_h}V_{h+1}(s) := r_h(s,a) + \sum_{s' \in \mathcal{S}} V_{h+1}(s')P^{\theta}_h(s'|s,\pi_h(s))$. Using Equation (6) of Osband et al. (2013), we have

$$\tilde{Z}_k = \mathbb{E}_{\theta^*,\tilde{\theta}^k}\left[\sum_{h=0}^{H-1}[\mathcal{T}^{\tilde{\theta}^k}_{\tilde{\pi}^k_h}V^{\tilde{\theta}^k}_{h+1}(s_{(k-1)H+h}) - \mathcal{T}^{\theta^*}_{\tilde{\pi}^k_h}V^{\tilde{\theta}^k}_{h+1}(s_{(k-1)H+h})]\right] \tag{23}$$

For convenience, we write $\tilde{P}^k_h = P^{\tilde{\theta}^k}_h, P^*_h = P^{\theta^*}_h, s_{k,h} = s_{(k-1)H+h}$, and $a_{k,h} = a_{(k-1)H+h} = \tilde{\pi}^k_h(s_{(k-1)H+h})$. Recall that the instantaneous reward satisfies $r_h(s,a) \in [0,1]$. We have

$$|\mathcal{T}^{\tilde{\theta}^k}_{\tilde{\pi}^k_h}V^{\tilde{\theta}^k}_{h+1}(s_{k,h}) - \mathcal{T}^{\theta^*}_{\tilde{\pi}^k_h}V^{\tilde{\theta}^k}_{h+1}(s_{k,h})| \leq H\|\tilde{P}^k_h(\cdot|s_{k,h},a_{k,h}) - P^*_h(\cdot|s_{k,h},a_{k,h})\|_1$$

Therefore,

$$\mathbb{E}[\tilde{Z}_k^2] \leq H\mathbb{E}\left[\sum_{h=0}^{H-1}[\mathcal{T}^{\tilde{\theta}^k}_{\tilde{\pi}^k_h}V^{\tilde{\theta}^k}_{h+1}(s_{k,h}) - \mathcal{T}^{\theta^*}_{\tilde{\pi}^k_h}V^{\tilde{\theta}^k}_{h+1}(s_{k,h})]^2\right] \tag{24}$$

$$\leq H^3\mathbb{E}\left[\sum_{h=0}^{H-1}\|\tilde{P}^k_h(\cdot|s_{k,h},a_{k,h}) - P^*_h(\cdot|s_{k,h},a_{k,h})\|_1^2\right] \tag{25}$$

where we used Cauchy-Schwartz inequality for the first inequality.

Following Osband et al. (2013), define $N_k(s,a,h) = \sum_{l=1}^{k-1}\mathbf{1}_{\{(s_{l,h},a_{l,h})=(s,a)\}}$ to be the number of times $(s,a)$ was sampled at step $h$ in the first $(k-1)$ episodes.

**Claim:** For any $k, h$ and any $\delta > 0$,

$$\mathbb{E}[\|\tilde{P}^k_h(\cdot|s_{k,h},a_{k,h}) - P^*_h(\cdot|s_{k,h},a_{k,h})\|_1^2] \leq 4\mathbb{E}\left[\frac{(2\log 2)S + 2\log(1/\delta)}{\max\{N_k(s_{k,h},a_{k,h},h),1\}}\right] + 8(k-1)SA\delta \tag{26}$$

Given the claim, we have

$$\sum_{k=1}^{T} \mathbb{E}[\tilde{Z}_k^2] \tag{27}$$

$$\leq H^3 \left[ 4\mathbb{E}\left[ \sum_{k=1}^{T} \sum_{h=0}^{H-1} \frac{(2\log 2)S + 2\log(1/\delta)}{\max\{N_k(s_{k,h}, a_{k,h}, h), 1\}} \right] + \sum_{k=1}^{T} 8(k-1)SA\delta \right] \tag{28}$$

$$\leq 4H^3 \mathbb{E}\left[ \sum_{k=1}^{T} \sum_{h=0}^{H-1} \frac{(2\log 2)S + 2\log(1/\delta)}{\max\{N_k(s_{k,h}, a_{k,h}, h), 1\}} \right] + H^3(4T^2SA\delta) \tag{29}$$

$$= 8H^3[(\log 2)S + \log(1/\delta)]\mathbb{E}\left[ \sum_{k=1}^{T} \sum_{h=0}^{H-1} (\max\{N_k(s_{k,h}, a_{k,h}, h), 1\})^{-1} \right] + 4H^3SAT^2\delta \tag{30}$$

It remains to bound the expectation term in Eq. (30). We have

$$\sum_{k=1}^{T} \sum_{h=0}^{H-1} (\max\{N_k(s_{k,h}, a_{k,h}, h), 1\})^{-1} = \sum_{h=0}^{H-1} \sum_{s\in\mathcal{S}} \sum_{a\in\mathcal{A}} \sum_{j=0}^{N_T(s,a,h)-1} \frac{1}{\max\{j, 1\}} \tag{31}$$

$$\leq \sum_{h=0}^{H-1} \sum_{(s,a):N_T(s,a,h)>0} (2 + \log(N_T(s, a, h))) \leq HSA(2 + \log(T)) \tag{32}$$

Combining Eq. (30) and Eq. (32), setting $\delta = \frac{1}{T^2}$, we have

$$\sum_{k=1}^{T} \mathbb{E}[\tilde{Z}_k^2] \tag{33}$$

$$\leq 8H^4SA[(\log 2)S + \log(1/\delta)][2 + \log(L)] + 4H^3SAT^2\delta \tag{34}$$

$$= 8H^4SA[(\log 2)S + 2\log(T)][2 + \log(T)] + 4H^3SA \tag{35}$$

$$= O(H^4S^2A[1 + \log(T)]^2) \tag{36}$$

Combining Eq. (21)Eq. (22)Eq. (36), we conclude that

$$\mathfrak{BR}_T = O(\sqrt{\varepsilon H^4 S^2 AT} \log(1+T)) \tag{37}$$

*Proof of Claim.* Let $\check{P}_h^n(\cdot|s,a)$ be the empirical distribution of transitions after sampling $(s,a)$ at step $h$ exactly $n$ times. By the $L_1$ concentration inequality for empirical distributions Weissman et al. (2003), we have

$$\mathbb{P}_{\theta^*}(\|\check{P}_h^n(\cdot|s,a) - P_h^{\theta^*}(\cdot|s,a)\|_1 > \epsilon) \leq 2^S \exp\left(-\frac{1}{2}n\epsilon^2\right) \tag{38}$$

for any $\epsilon > 0$. For $\delta > 0$, define $\xi(n,\delta) := \frac{(2\log 2)S + 2\log(1/\delta)}{\max\{n,1\}}$, we have

$$\mathbb{P}_{\theta^*}\left(\|\check{P}_h^n(\cdot|s,a) - P_h^{\theta^*}(\cdot|s,a)\|_1^2 > \xi(n,\delta)\right) \leq \delta. \tag{39}$$

Define $\hat{P}_h^k(\cdot|s,a)$ to be the empirical distribution of transitions after sampling $(s,a)$ at step $h$ in the first $k-1$ episodes. Define

$$\hat{\Theta}_h^k = \left\{\theta : \|\hat{P}_h^k(\cdot|s,a) - P_h^\theta(\cdot|s,a)\|_1^2 \leq \xi(N_k(s,a,h),\delta) \;\; \forall s\in\mathcal{S}, a\in\mathcal{A}\right\}$$

Then, we have

$$\mathbb{P}(\theta^* \notin \hat{\Theta}_h^k) \tag{40}$$

$$\leq \mathbb{P}\left(\exists n \in [k-1], s \in \mathcal{S}, a \in \mathcal{A} \ \ \|\check{P}_h^n(\cdot|s,a) - P_h^{\theta^*}(\cdot|s,a)\|_1^2 > \xi(n,\delta)\right) \tag{41}$$

$$\leq (k-1)SA\delta \tag{42}$$

Since $\theta^*$ and $\tilde{\theta}^k$ are i.i.d. given $\mathcal{D}_k$, and the random set $\hat{\Theta}_h^k$ is measurable to $\mathcal{D}_k$, we have $\mathbb{P}(\theta^* \notin \hat{\Theta}_h^k) = \mathbb{P}(\tilde{\theta}^k \notin \hat{\Theta}_h^k)$. Therefore we have

$$\mathbb{E}\left[\|\tilde{P}_h^k(\cdot|s_{k,h},a_{k,h}) - P_h^*(\cdot|s_{k,h},a_{k,h})\|_1^2\right] \tag{43}$$

$$\leq \mathbb{E}\left[\|\tilde{P}_h^k(\cdot|s_{k,h},a_{k,h}) - P_h^*(\cdot|s_{k,h},a_{k,h})\|_1^2 \mathbf{1}_{\{\theta^*,\tilde{\theta}^k \in \hat{\Theta}_h^k\}}\right] + \tag{44}$$

$$+ \mathbb{E}\left[\|\tilde{P}_h^k(\cdot|s_{k,h},a_{k,h}) - P_h^*(\cdot|s_{k,h},a_{k,h})\|_1^2 \left(\mathbf{1}_{\{\theta^* \notin \hat{\Theta}_h^k\}} + \mathbf{1}_{\{\tilde{\theta}^k \notin \hat{\Theta}_h^k\}}\right)\right] \tag{45}$$

$$\leq \mathbb{E}[4\xi(N_k(s_{k,h},a_{k,h},h),\delta)] + 4\left(\mathbb{P}(\theta^* \notin \hat{\Theta}_h^k) + \mathbb{P}(\tilde{\theta}^k \notin \hat{\Theta}_h^k)\right) \tag{46}$$

$$\leq 4\mathbb{E}[\xi(N_k(s_{k,h},a_{k,h},h),\delta)] + 8(k-1)SA\delta \tag{47}$$

$$\square$$

## C   Proof of Lemma A.2

*Proof.* For convenience, write $\mathbb{P}_\theta(\cdot) = \mathbb{P}(\cdot|\theta)$.

Define the event $\mathcal{E}_{n,h} = \{\bar{a}_{n,h} \neq a_h^*(\bar{s}_{n,h};\theta)\}$, i.e. in the $n$-th round of demonstration, the expert did not take the optimal action at time $h$. Given the expert's randomized policy $\phi$, we have

$$\mathbb{P}_\theta(\mathcal{E}_{n,h}|\bar{s}_{n,0:h},\bar{a}_{n,0:h-1}) \tag{48}$$

$$= 1 - \frac{1}{1 + \sum_{a \neq a_h^*(\bar{s}_{n,h};\theta)} \exp(-\beta\Delta_h(\bar{s}_{n,h},a;\theta))} \tag{49}$$

$$\leq \sum_{a \neq a_h^*(\bar{s}_{n,h};\theta)} \exp(-\beta\Delta_h(\bar{s}_{n,h},a;\theta)) \tag{50}$$

$$\leq (A-1)\exp(-\beta\Delta) =: \tilde{\kappa}_\beta. \tag{51}$$

Define $\kappa_\beta = (H-1)\tilde{\kappa}_\beta$. Then $\beta \geq \underline{\beta}$ means that $\kappa_\beta \leq \underline{p}/3$.

Consider each $(h,s) \in \{0,1,\cdots,H-1\} \times \mathcal{S}$, conditioning on $\theta$ there are two cases:

- If $p_h(s;\theta) > 0$ then

$$\mathbb{P}_\theta(\bar{s}_{n,h} = s) \geq (1-\tilde{\kappa}_\beta)^h p_h(s;\theta) \geq (1-\kappa_\beta)\underline{p} \geq \frac{2}{3}\underline{p}. \tag{52}$$

  The first inequality in Eq. (52) can be established via induction on $h$: First observe that $\mathbb{P}_\theta(\bar{s}_{n,0} = s) = p_0(s;\theta)$ for all $s \in \mathcal{S}$ by definition. Suppose that we have proved the statement for time $h$, i.e. $\mathbb{P}_\theta(\bar{s}_{n,h} = s) \geq (1-\tilde{\kappa}_\beta)^h p_h(s;\theta)$ for all $s \in \mathcal{S}$. Then we have

$$p_{h+1}(s';\theta) = \sum_{s \in \mathcal{S}} \mathbb{P}_\theta(s'|s,a_h^*(s;\theta))p_h(s;\theta) \tag{53}$$

$$\mathbb{P}_\theta(\bar{s}_{n,h+1} = s') \geq \sum_{s \in \mathcal{S}} \phi_h^\beta(a_h^*(s;\theta)|s;\theta)\mathbb{P}_\theta(s'|s,a_h^*(s;\theta))\mathbb{P}_\theta(\bar{s}_{n,h} = s) \tag{54}$$

$$\geq \sum_{s \in \mathcal{S}} (1-\tilde{\kappa}_\beta)\mathbb{P}_\theta(s'|s,a_h^*(s;\theta))\mathbb{P}_\theta(\bar{s}_{n,h} = s) \tag{55}$$

$$\geq \sum_{s \in \mathcal{S}} (1-\tilde{\kappa}_\beta)^{h+1}\mathbb{P}_\theta(s'|s,a_h^*(s;\theta))p_h(s;\theta). \tag{56}$$

The statement for $h+1$ then follows by comparing Eq. (53) and Eq. (56), establishing the induction step.

- If $p_h(s;\theta) = 0$, then if $\bar{s}_{n,h} = s$, the expert must have chosen some action that was not optimal before time $h$ in the $n$-th round of demonstration. We conclude that

$$\mathbb{P}_\theta(\bar{s}_{n,h} = s) \leq \mathbb{P}_\theta\left(\bigcup_{\tilde{h}=1}^{h-1} \mathcal{E}_{n,\tilde{h}}\right) \leq \sum_{\tilde{h}=1}^{h-1} \mathbb{P}_\theta(\mathcal{E}_{n,\tilde{h}}) \leq \kappa_\beta \leq \frac{1}{3}\underline{p}. \tag{57}$$

The above argument shows that there's a separation of probability between two types of state and time index pairs under the expert's policy $\phi^\beta(\theta)$: the ones that are probable under the optimal policy $\pi^*(\theta)$ and the ones that are not. Using this separation, we will proceed to show that when $L$ is large, we can distinguish the two types of state and time index pairs through their appearance counts in $\mathcal{D}_0$. This will allow us to construct a good estimator of $\pi^*$.

Define $\hat{\pi}^*$ to be the estimator of $\pi^*$ constructed with $\delta = \underline{p}/2$. If $\hat{\pi}^* \neq \pi$, then either one of the following cases happens

- There exists an $(s,h) \in \mathcal{S} \times \{0, 1, \cdots, H-1\}$ pair such that $p_h(s;\theta) > 0$ but $N_h(s) < \delta L$;

- There exists an $(s,h) \in \mathcal{S} \times \{0, 1, \cdots, H-1\}$ pair such that $p_h(s;\theta) = 0$ but $N_h(s) \geq \delta L$;

- There exists an $(s,h) \in \mathcal{S} \times \{0, 1, \cdots, H-1\}$ pair such that $p_h(s;\theta) > 0$ and $N_h(s) \geq \delta L$, but $\pi_h^*(s;\theta) = a_h^*(s;\theta) \neq \arg\max_a N_h(s,a) = \hat{\pi}_h^*(s)$;

Using union bound, we have

$$\mathbb{P}_\theta(\pi^* \neq \hat{\pi}^*) \tag{58}$$

$$\leq \sum_{(s,h):p_h(s;\theta)>0} \mathbb{P}_\theta(N_h(s) < \delta L) + \sum_{(s,h):p_h(s;\theta)=0} \mathbb{P}_\theta(N_h(s) \geq \delta L) \tag{59}$$

$$+ \sum_{(s,h):p_h(s;\theta)>0} \mathbb{P}_\theta(N_h(s) \geq \delta L, a_h^*(s;\theta) \neq \arg\max_a N_h(s,a)). \tag{60}$$

Let $\text{Bin}(M, q)$ denote a binomial random variable with parameters $M \in \mathbb{N}$ and $q \in [0,1]$. Notice that conditioning on $\theta$, each $N_h(s)$ is a binomial random variable with parameters $L$ and $\tilde{p}_{\theta,h}(s) := \mathbb{P}_\theta(\bar{s}_{1,h} = s)$.

Using equation 52 and Lemma A.1, we conclude that each term in the first summation of equation 60 satisfies

$$\mathbb{P}_\theta(N_h(s) < \delta L) \leq \mathbb{P}(\text{Bin}(L, 2\underline{p}/3) < (\underline{p}/2)L) \tag{61}$$

$$\leq \exp(-2L(\underline{p}/6)^2) = \exp\left(-\frac{L\underline{p}^2}{18}\right). \tag{62}$$

Using equation 57 and Lemma A.1, we conclude that each term in the second summation of equation 60 satisfies

$$\mathbb{P}_\theta(N_h(s) \geq \delta L) \leq \mathbb{P}(\text{Bin}(L, \kappa_\beta) \leq (\underline{p}/2)L) \tag{63}$$

$$\leq \exp\left(-2L\left(\frac{\underline{p}}{2} - \kappa_\beta\right)^2\right) \leq \exp\left(-\frac{L\underline{p}^2}{18}\right). \tag{64}$$

Again, using Lemma A.1, each term in the third summation of equation 60 satisfies

$$\mathbb{P}_\theta(N_h(s) \geq \delta L, a_h^*(s;\theta) \neq \arg\max_a N_h(s,a)) \tag{65}$$

$$\leq \mathbb{P}_\theta(a_h^*(s;\theta) \neq \arg\max_a N_h(s,a) \mid N_h(s) \geq \delta L) \tag{66}$$

$$\leq \mathbb{P}_\theta(N_h(s) - N_h(s,a_h^*(s;\theta)) \geq N_h(s)/2 \mid N_h(s) \geq \delta L) \tag{67}$$

$$\leq \mathbb{P}_\theta(\mathrm{Bin}(N_h(s),\tilde{\kappa}_\beta) \geq N_h(s)/2 \mid N_h(s) \geq \delta L) \tag{68}$$

$$\leq \mathbb{P}_\theta(\mathrm{Bin}(N_h(s),1/3) \geq N_h(s)/2 \mid N_h(s) \geq \delta L) \tag{69}$$

$$\leq \mathbb{E}_\theta\left[\exp\left(-\frac{N_h(s)}{18}\right) \,\Big|\, N_h(s) \geq \delta L\right] \tag{70}$$

$$\leq \exp\left(-\frac{\delta L}{18}\right) = \exp\left(-\frac{L\underline{p}}{36}\right). \tag{71}$$

Combining the above we obtain

$$\mathbb{P}_\theta(\pi^* \notin \hat{\pi}^*) \leq SH\left[\exp\left(-\frac{L\underline{p}^2}{18}\right) + \exp\left(-\frac{L\underline{p}}{36}\right)\right]. \tag{72}$$

$\square$

## D  Empirical Results

### D.1  Pseudo-codes for RLSVI agents

In this appendix, we provide the pseudo-codes for `iRLSVI`, `piRLSVI`, and `uRLSVI` used in experiments in Section 5, which are illustrated in Algorithm 2 and 3. Note that all three agents are tabular RLSVI agents with similar posterior sampling-type exploration schemes. However, they differ in whether or not and how to exploit the offline dataset. More precisely, they differ in use of the offline dataset $\mathcal{D}_0$ when computing the RLSVI loss $\tilde{\mathcal{L}}_{\mathrm{RLSVI}}$ (see Algorithm 2 and 3), and if the total loss includes the loss associated with the expert actions $\tilde{\mathcal{L}}_{\mathrm{IR}}$ (see Algorithm 3). Table 1 summarizes the differences. In all experiments in this paper, we choose the algorithm parameters $\sigma_0^2 = 1$, $\sigma^2 = 0.1$, and $B = 20$.

| agent | use $\mathcal{D}_0$ in $\tilde{\mathcal{L}}_{\mathrm{RLSVI}}$? | include $\tilde{\mathcal{L}}_{\mathrm{IR}}$? |
|---|---|---|
| `uRLSVI` | No | No |
| `piRLSVI` | Yes | No |
| `iRLSVI` | Yes | Yes |

Table 1: Differences between `iRLSVI`, `piRLSVI`, and `uRLSVI`.

### D.2  More empirical results on Deep Sea

In this appendix, we provide more empirical results for the Deep Sea experiment described in Section 5. Specifically, for Deep Sea with size $M = 10$, data ratio $\kappa = 1, 5$, and expert's deliberateness $\beta = 1, 10$, we plot the cumulative regret of `iRLSVI`, `piRLSVI`, and `uRLSVI` as a function of the number of episodes $t$ for the first $T = 300$ episodes. The experiment results are averaged over 50 simulations and are illustrated in Figure 5.

As we have discussed in the main body of the paper, when both data ratio $\kappa$ and the expert's deliberateness $\beta$ are small, then there are not many offline data and the expert's generative policy is also not very informative. In this case, `iRLSVI`, `piRLSVI`, and `uRLSVI` perform similarly. On the other hand, when the data ratio $\kappa$ is large, `iRLSVI` and `piRLSVI` tend to perform much better than `uRLSVI`, which does not use the offline dataset. Similarly, when the expert's deliberateness $\beta$ is large, then the expert's generative policy is informative. In this case, `iRLSVI` performs much better than `piRLSVI` and `uRLSVI`.

---

**Algorithm 2** RLSVI agents for numerical experiments

---

**Input:** algorithm parameter $\sigma_0^2, \sigma^2 > 0$, deliberateness parameter $\beta > 0$, offline dataset $\mathcal{D}_0$, offline buffer size $B$, agent type `agent`

**if** `agent` = `uRLSVI` **then**
    initialize data buffer $\mathcal{D}$ as an empty set
**else**
    initialize data buffer $\mathcal{D} \leftarrow \mathcal{D}_0$
**end if**

**for** $t = 1, \cdots, T$ **do**
    sample state-action value function $\hat{Q}^t$ backwardly based on Algorithm 3
    sample initial state $s_0^t \sim \nu$
    **for** $h = 0, \cdots, H - 1$ **do**
        take action $a_h^t \sim \text{Unif}\left(\arg\max_a \hat{Q}_h^t(s_h^t, a)\right)$
        observe $(s_{h+1}^t, r_h^t)$
        append $(s_h^t, a_h^t, h, s_{h+1}^t, r_h^t)$ to the data buffer $\mathcal{D}$
    **end for**
**end for**

---

**Algorithm 3** sample $\hat{Q}^t$

---

**Input:** algorithm parameter $\sigma_0^2, \sigma^2 > 0$, deliberateness parameter $\beta > 0$, offline dataset $\mathcal{D}_0$, data buffer $\mathcal{D}$, offline buffer size $B$, agent type `agent`

set $\hat{Q}_{H+1}^t \leftarrow 0$

**for** $h = H, H - 1, \cdots, 0$ **do**
    Let $Q \in \Re^{|\mathcal{S}| \times |\mathcal{A}|}$ denote the decision variable, then we define the RLSVI loss function

$$\tilde{\mathcal{L}}_{\text{RLSVI}}(Q) = \frac{1}{2} \sum_{d=(s,a,h',s',r) \in \mathcal{D}} \left( Q(s,a) - (r + \omega_d) - \max_b \hat{Q}_{h+1}^t(s',b) \right)^2 + \frac{1}{2\sigma_0^2} \|Q - Q^{\text{prior}}\|_{\text{F}}^2,$$

    where for each data entry $d = (s, a, s', r) \in \mathcal{D}$, $\omega_d$ is i.i.d. sampled from $N(0, \sigma^2)$. Each element in $Q^{\text{prior}} \in \Re^{|\mathcal{S}| \times |\mathcal{A}|}$ is i.i.d. sampled from $N(0, \sigma_0^2)$ and $\| \cdot \|_F$ denotes the Frobenius norm.
    **if** `agent` = `iRLSVI` **then**
        sample a buffer of $B$ offline data tuple, $\mathcal{B}$, without replacement from the offline dataset $\mathcal{D}_0$
        define $\mathcal{B}_h = \{(s, a, h', s, , r) \in \mathcal{B} : h' = h\}$
        define the loss function associated with expert actions as

$$\tilde{\mathcal{L}}_{\text{IL}}(Q) = \sum_{(s,a,h',s',r) \in \mathcal{B}_h} \left[ \log \sum_b \exp\left(\beta Q(s,b)\right) - \beta Q(s,a) \right]$$

        choose $\hat{Q}_h^t \in \arg\min_Q \left[ \tilde{\mathcal{L}}_{\text{RLSVI}}(Q) + \tilde{\mathcal{L}}_{\text{IL}}(Q) \right]$
    **else**
        choose $\hat{Q}_h^t \in \arg\min_Q \tilde{\mathcal{L}}_{\text{RLSVI}}(Q)$
    **end if**
**end for**
**Return** $\hat{Q}^t = \left( \hat{Q}_0^t, \hat{Q}_1^t, \ldots, \hat{Q}_H^t \right)$

---

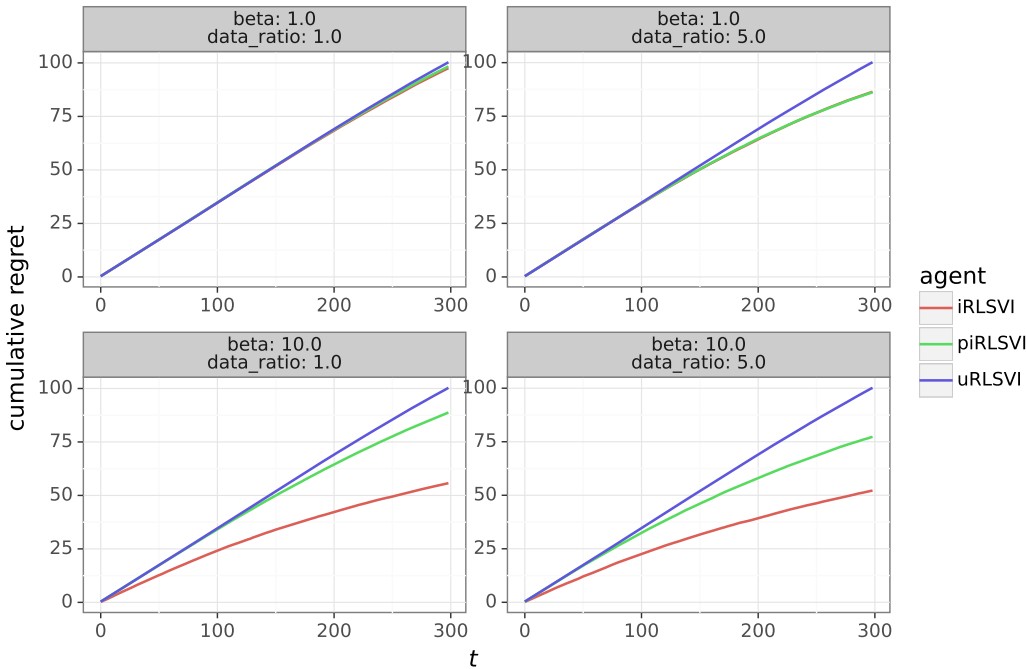

Figure 5: Cumulative regret vs. number of episodes in Deep Sea.

## D.3    More empirical results on Maze

In this appendix, we provide more empirical results for the Maze experiment described in Section 5. Specifically, for the Maze environment, data ratio $\kappa = 1, 5$, and expert's deliberateness $\beta = 1, 10$, we plot the cumulative regret of `iRLSVI`, `piRLSVI`, and `uRLSVI` as a function of the number of episodes $t$ for the first $T = 200$ episodes. The experiment results are averaged over 50 simulations and are illustrated in Figure 6. We have observed similar experiment results as the Deep Sea experiment.

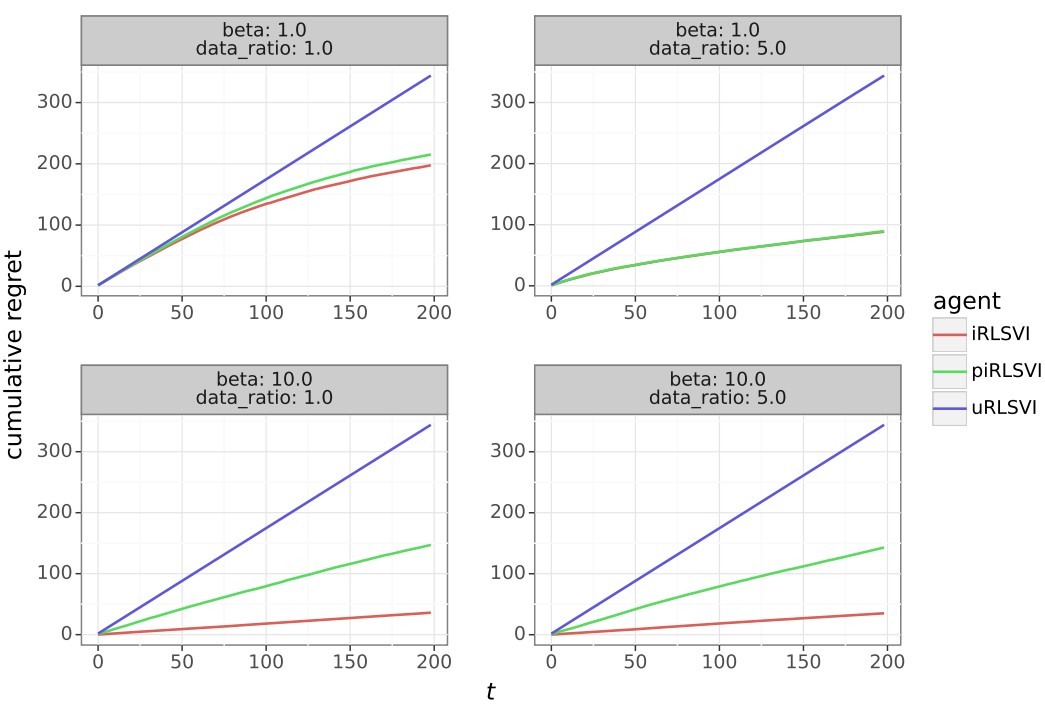

Figure 6: Cumulative regret vs. number of episodes in Maze.

