# OpenReview forum: "Bridging Imitation and Online Reinforcement Learning: An Optimistic Tale"
_TMLR — Accepted by TMLR_

### Review · Reviewer_UsQL · 2023-03-20

**Summary Of Contributions:**

The paper studies the problem of how one leverages the offline demonstration dataset from an imperfect expert to improve online learning. The authors propose Informed Posterior Sampling-based RL (iPSRL), which incorporates the prior information learned from offline demonstration into the existing PSRL algorithm. Experimental results show that the proposed algorithm improves over the partially informed and uninformed PSRL algorithms.


**Audience:**

Yes

**Claims And Evidence:**

Yes

**Requested Changes:**

I think the contribution of the work is not enough for publication on TMLR. However, I think the following change will significantly improve the contribution:
1. Include lower bounds for both algorithms, and provide analysis to improve the exponential dependency of H as the authors mentioned in the remark.
2. Add more theoretical and empirical results that compare the proposed algorithm with the standard baseline, which first trains a policy and value function from the offline demonstration dataset, and then explores based on it using Q-learning, SAC, or policy-based algorithms.
3. Add empirical results on D4RL dataset.


**Strengths And Weaknesses:**

Strength:

1. The paper provides a regret guarantee for both PSRL and iPSRL, which shows how the prior information in offline demonstration dataset can help improve the regret.
2. The paper also provides an approximate algorithm that can be efficiently implemented.
3. The paper interprets the new loss as a combination of the loss for online learning and that of imitation learning.

Weakness:

1. I’m confused how Lemma 3.1 is related to the results in the prior work, since PSRL is a standard algorithm with a good amount of theoretical analysis. A more detailed discussion on related work is needed here.
2. It’s unclear how large the entropy term can be in both Lemma 3.1 and Corollary 3.2. Since the only improvement is in the entropy term, it’s better to have some examples to see when the first term in the minimum can be smaller than the second term in the regret bound of Corollary 3.2.
3. It’s better to see some discussions on the tightness of Lemma 3.1 and Corollary 3.2. Otherwise one cannot say that iPSRL improves over PSRL since the lower bound for PSRL is missing.
4. I wonder how the proposed algorithm compares with the simplest baseline: one first trains a policy purely from offline demonstration dataset using imitation learning (and also learn a Q function using max entropy RL), then explores using Q-learning, SAC, or policy-based algorithms. It would be great to see theoretical and empirical comparisons. I understand that the Bayesian algorithm is usually not comparable in theory with existing value-based algorithms since the assumptions are quite different. But an empirical study would be greatly appreciated.  Since such a baseline is the most common practice in the real world, it would be great if the authors can demonstrate the superiority of the proposed algorithm over it. It is also better to see experiments on practical datasets like D4RL.

---

> ### Author Response · Authors · 2023-05-03
> **Thanks for your comments and response**
>
> 1.     Lemma 3.1 in relation to prior work:  We have added new results (Lemmas 3.1 & 3.2 and Theorem 3.7) such that exponential dependency of H is removed. When the offline data is not informative, our regret bound $O(\sqrt{H^4S^2AL})$ nearly matches the minimax lower bound $O(\sqrt{H^2SAL})$ (Azar, et al ICML 2017) up to $H^2S$ factors. Obtaining the exact minimax optimality for Thompson sampling in MDPs is still an open problem and requires much involved further analysis.
>
> 2.     Entropy term in Lemma 3.1 and Corollary 3.2: We have added a new proof that does not rely on those entropy terms now.
>
> 3.     Tightness of Lemma 3.1 and Corollary 3.2: We do not need the old Lemma 3.1 and Corollary 3.2 anymore. They are replaced by new Lemmas 3.1 and 3.2.
>
> 4.     Comparison to a simple baseline: We note that it is well-understood in the literature that the simple baseline(s) mentioned by the reviewer would fail to have sublinear regret because they do not do deep exploration (see Osband, et al (2018) for a discussion) that problems like Deep Sea environment require. Moreover, it is not clear how to incorporate offline datasets in their training, and even less clear how to incorporate any information about the behavioral policy in these algorithms. Such a comparison in any case would not be meaningful because these algorithms are not designed for efficient exploration to achieve regret minimization. We do plan to add comparison to Q-learning with UCB Hoeffding [Jin et al (2018)] in the next version.
>
> 5. Empirical results on D4RL environments. We note that D4RL environments are mostly continuous/very large state and action space problems. That requires generalization through function approximations. This would be a useful future work but beyond the scope of this paper. Our intent here is to provide a proof-of-concept of the ideas in tabular problem where deep exploration is needed, and Deep Sea is definitely the benchmark problem for it.

---

### Review · Reviewer_TwXf · 2023-04-07

**Summary Of Contributions:**

The authors introduce a method for learning from an offline dataset while learning online. They first describe a method with unrealistic assumptions but strong regret guarantees, then introduce a more practical approximation. The agent is tested in the Deep Sea environment.

**Audience:**

Yes

**Broader Impact Concerns:**

No concerns. A broader impact statement is not needed.

**Claims And Evidence:**

No

**Requested Changes:**

See listed weaknesses.

**Strengths And Weaknesses:**

The writing in this paper is unclear and leaves out a lot of relevant and important details. It is often unclear whether contributions are novel or imported from prior work. This makes it difficult to judge both the strengths and weaknesses of the paper. This type of paper is not my area, so consider my confidence to be low.

- (Weakness) The claim that this is the first paper to bridge online RL and imitation learning seems very overstated. There are many papers in the Learning from Demonstrations literature (LfD) which do so. There are many papers in the offline RL literature which combine RL and imitation.

- (Weakness) The claim that "we have introduced and studied a new problem: Given an offline demonstration dataset from an imperfect expert, what is the best way to leverage it to bootstrap online learning performance in MDPs" seems very overstated. It's unclear to me, if the authors are implying that this is distinct from the offline finetuning setting because there is no offline pre-training, but this seems pedantic at best, as many of these methods are capable, but pre-train as this is logical to do when given offline data.

- (Weakness) The paper seems heavily based on PSRL and RLSVI but are not described or described not in detail. Citations for RLSVI point to several distinct papers, and seemingly (based on the loss function in 4.2) do not refer to the "least squares" (the LS in RLSVI) version. This makes it hard to understand what the contributions of the paper are.

- (Weakness) Algorithmic details are missing or out of order. Corollary 3.2 refers to the regret of the iPSRL algorithm, which is only described by eqn (3). There is no description of how the policy is determined. Its possible the policy of iPSRL is determined on page 7, but this is never stated clearly. iRLSVI seems to make use of Remark 4.1/4.2 but any exact details are omitted in the paper.

- (Weakness) Remark 4.1 suggests the authors estimate $\beta$ by considering the entropy of the dataset. In practice this assumption is completely unrealistic, as the behavior policy could be deterministic, or the expert could be very stochastic.

- (Weakness) The experimental results are unsurprising/meaningless because they do not compare against any competitive baselines. It is obvious that giving more data/more information would improve results. The authors should compare with RLVSI with offline data (and without their adjustments) as well as other offline RL methods, particularly ones used for offline+finetuning.

- (Weakness) There are missing details for reproducibility of the experimental results when regarding the algorithmic details of the method and baselines. I realize this is not the main contribution of the paper, but these details are easy to include and there's no reason to hold theoretically grounded papers to a lower standard on reproducibility.

- (Strength) As a tool for analysis (both theoretical and empirical), the formulation of the expert is a useful contribution, as it is more meaningful then just expert + some noise level.

- (Strength) Theoretical contributions seem interesting but it is unclear to me what is novel over PSRL and RLSVI due to clarity challenges. Hopefully this will be more clear with revisions.

Minor:

- In introduction: "This often suffers from the sim2real problem", sim2real refers to the problem from moving from simulation to real world environments. This should be "distribution shift" or similar terms since the authors refer to the challenge from moving from offline to online.
- PSRL/RLSVI are motivated heavily by exploration/optimism, it was unclear to me how that motivation played into the variants proposed in this paper.

---

> ### Author Response · Authors · 2023-05-03
> **Thanks for your comments and response**
>
> We address the weakness mentioned by the reviewers below.
> 1.     Claim about bridging Online RL and Imitation Learning: We acknowledge that there is a vast literature on Imitation Learning (or LfD) as well as Offline RL. There are even papers on Offline RL with online policy fine-tuning. But these works focus on policy optimization, i.e., find a near-optimal policy w.h.p. and not regret minimization, a key metric for online learning efficiency. The closest paper to ours is [Rashidinejad, et al (2021)] which “bridges” offline RL and imitation learning. But note that our problem formulation is quite different (regret-minimization during the online phase). Moreover, our algorithm is able to work well (in terms of sublinear regret) even if the offline dataset is not of good quality, as characterized by  concentrability coefficients as introduced in the offline RL literature, and as used in [Rashidinejad, et al (2021)]. Nevertheless, we have toned down the scope of our claims and made them more specific.
> 2.     Comparison to Offline RL+ Online Finetuning: As explain in 1), while the offline RL + online finetuning problem has a similar motivation, the problem objectives are quite different, i.e., policy optimization v. regret minimization. Regret minimization is a suitable objective for online learning because it appropriately captures the need to balance exploration v. exploitation, and is an accepted measure learning efficiency in online learning settings. We believe ours is the first paper that does this when an offline dataset is available to start with. If the reviewer knows of any reference we have missed, could you please point it out to us. We will include that also.
> 3.     Algorithms’ pseudo-code: PSRL and RLSVI are standard algorithms in the Online RL literature. So we just provide citations to suitable papers. We have now included pseudo-code for iPSRL and iRLSVI so the reader can clearly understand the algorithm.
> 4.     Estimating beta from the dataset: Note that under our generative model (2), if the expert policy is very stochastic, this expert can be far from near-optimal. In this case, the entropy estimation is large, which suggests the regularization over the behavior policy is small.
> 5.     Comparison to competitive baselines: Please note that, while there are some algorithms for offline RL + online finetuning, their objective is policy optimization. We believe  comparison against such algorithms will not be fair as it is well-known in Online RL literature that such algorithms would perform poorly in environments such as DeepSea that require deep exploration. We plan to add comparison to Q-learning with UCB Hoeffding [Jin et al (2018)] in the next version.  If the author knows of any other algorithm for online RL with use of offline dataset with regret minimization as an objective, could you please point it out to us?
>
> 6.      Reproducibility of experimental results: This is a good point, and we have included more details in the paper to enable reproducibility of the experimental results.

---

### Review · Reviewer_scp5 · 2023-04-13

**Summary Of Contributions:**

This work brings the posterior sampling reinforcement learning (PSRL) and randomized least-squares value iteration (RLSVI) algorithms to the online fine-tuning setting, where the agent has access to a dataset of trajectories collected *a priori* by a behavior policy. It proposes “informed” variants iPSRL and iRLSVI, which leverage this offline dataset and knowledge of the behavior policy. These algorithms show clear benefits over the original algorithms in this setting.

**Audience:**

Yes

**Broader Impact Concerns:**

None.

**Claims And Evidence:**

No

**Requested Changes:**

The claims about this work being the first to bridge online and offline RL, or online RL and imitation learning, need to either be removed or made substantially more specific.

The second paragraph of the introduction says that offline RL suffers from a “sim2real” problem. I think this was meant to be “distribution shift” or “compounding errors”.

The notation is quite dense and slow to read. I wonder if the main body of the manuscript might be made clearer by simpler and less precise notation, reserving multiple super/subscripts (e.g. the tuple $(s_h^l, \tilde{a}_h^l, \check{s}_h^l, \tilde{\mathcal{E}}_h^l)$, page 7) for the appendix where possible. It’s fine if this is not feasible.

**Strengths And Weaknesses:**

### Strengths

The proposed algorithms are interesting and consist of intuitive modifications to their online counterparts.

The final algorithms in Eqs. 10 and 11 mirror what has been seen in the empirical literature, and their derivations provide another framework for understanding the success of existing work.

The theory results look reasonable upon a light reading, though the relationship to PSRL and RLSVI is not made explicit.

### Weaknesses

The paper makes claims in multiple places about being the first work that addresses this setting. In some places those claims are more tightly constrained than in others, but some versions of the statement are patently false. The abstract is perhaps the worst offender, stating that “Our algorithm bridges online RL and imitation learning for the first time.” There are a variety of algorithms in the deep RL community which do this in different ways; see [1] for a particularly well-known example of work which does this algorithmically, or [2] for one which studies this setting. A similar claim appears at the end of the introduction.

It was not precisely clear to me what information the agent had about the behavior policy in each of these settings and where that information was used. Does the agent know fully $(\beta, \lambda)$? Just $\beta$? In the first paragraph of section 3, it notes that $\beta$ is known to the expert — was that meant to be, known to the agent?

[1]: Fujimoto, S., & Gu, S.S. (2021). A Minimalist Approach to Offline Reinforcement Learning. *ArXiv, abs/2106.06860*.

[2]: Kostrikov, I., Nair, A., & Levine, S. (2021). Offline Reinforcement Learning with Implicit Q-Learning. *ArXiv, abs/2110.06169*.

---

> ### Author Response · Authors · 2023-05-03
> **Thanks for your comments and response**
>
> 1. Claim about bridging Online RL and Imitation Learning: We thank the reviewer for the valuable feedback. We first want to mention that in the special case that there is no offline dataset, iPSRL and iRLSVI algorithms reduce to the well-known PSRL and RLSVI algorithms for episodic MDPs respectively. We acknowledge that there are a number of papers that have proposed algorithms for offline RL with online finetuning, but what we specifically mean is that ours is the first formulation for design of regret-minimizing online learning algorithms that also make use of offline datasets. While we are aware of paper [Rashidinejad, et al (2021)] which bridges offline RL and imitation learning, we are unaware of any other work that does the same for Online RL and Imitation learning. We believe prior to this paper, information about the behavioral policy is not incorporated into the online learning agent’s policy. Nevertheless, we have toned down the language at various places in the paper and made the claims more specific.
>
> 2. Sim2real has been changed to distribution shift.
> 3. Readability: We have fixed the other issues, and made the paper easier to read without losing mathematical precision, and since another reviewer actually seems to be asking for more details, not less.
>
> [3] Paria Rashidinejad, Banghua Zhu, Cong Ma, Jiantao Jiao, and Stuart Russell. Bridging offline reinforcement learning and imitation learning: A tale of pessimism. Advances in Neural Information Processing Systems, 34:11702–11716, 2021

---

### Author Response · Authors · 2023-05-03
**Revised manuscript to address reviewers' comments**

We would like to thank the reviewers for a detailed feedback. Significant changes in the revised manuscript are:
(i) We have revised the entire manuscript as per reviewer feedback.
(ii) Regret bound for the iPSRL algorithm has been improved to now have only poly(H) dependence. Thus, Lemmas 3.1 and 3.2 and Theorem 3.7 are new. Proofs are updated/available  in Appendix B.
(iii) Pseudocode for the various algorithms is also added both in the main papers and in Appendix D.
(iv) Major changes are in BLUE text.

Please note that we are still assessing if any meaningful comparison can be made to any other relevant algorithms. If so, we will include further empirical results in the next revision.

---

### Decision · Action_Editors · 2023-06-08

**Recommendation:** Accept with minor revision

**Comment:**

While the theoretical analysis is good, the empirical analysis is a bit weak, i.e., with only an oversimplified game, tabular case, without baselines. The authors are suggested to enhance empirical studies, e.g.,
- Conduct experiments on D4RL environments. While theoretical analysis with function approximation indeed needs more effort, it will greatly strengthen this paper. Furthermore, even if without theoretical analysis, it is also valuable to conduct experiments over this more challenging setting (continuous/very large state and action space problems)
- Or conduct experiments on several other tabular environments. Although those environments might not be as valuable as D4RL environments, they will still enhance the empirical studies in the paper.

**Audience:**

Yes

**Claims And Evidence:**

- The theoretical analysis is good.
- The empirical analysis is a bit weak, i.e., with only an oversimplified game, tabular case, without baselines.